# Towards zero shot multivariate time series anomaly detection - A Realistic Evaluation

## Abstract

Multivariate time series anomaly detection (MTAD) approaches predominantly use performance enhancements that are not practical. E.g., a) point adjustment (PA) technique is employed which uses ground truth to forcefully convert false negatives to true positives and unrealistically inflates precision, and b) significant data leakage is incurred when anomaly score threshold is determined using the test data and test labels. This paper first presents real-world performance of existing MTAD techniques without PA and threshold learning (TL) on test data which shows anomalies in real-world benchmarks result in significant distribution shift between normal and anomalous data; and with PA and TL, even untrained deterministic methods can perform on par or even beat baseline techniques. Next it introduces six synthetic benchmarks based on real-world systems, where anomalous data and normal data have statistically almost same distributions. It then presents, sparse model identification enhanced anomaly detection (SPIE-AD), a novel model recovery and conformance based zero-shot MTAD approach that outperforms state-of-art MTAD on three real-world benchmarks without using PA and TL on test data. Extensive peformance results show that SPIE-AD outperforms SOTA MTAD techniques on both standard and novel benchmarks.

## 1 Introduction

Time series anomaly detection is essential for a safe and effective operation of unmanned aerial vehicles (UAV), autonomous cars (AC), and autonomous drug delivery (ADD) systems due to a complex amalgamation of interacting perception, decision making and actuations. Such complexity makes testing for "all possible" operational scenarios practically infeasible. Test cases ignored during pre-deployment evaluation but that occur during deployment, called "unknown unknowns" (U2), are a major cause of accidents (Maity et al., 2023). U2 detection is a special case of zero-shot anomaly (ZSA) detection, when anomaly data is unavailable during model training. We present **SPIE-AD**, **SP**arse model **I**dentification **E**nhanced **A**nomaly **D**etection which continually mines underlying sparse physical dynamics and checks its conformance with the original lab tested system.

U2s can potentially occur due to: a) **hardware changes/failures**, which may not be monitored, e.g. mechanical failure in an aircraft resulting in an elevator getting stuck ($F8Stuck$) or moving slow ($F8Slow$), b) **unwanted software executions:** which may not immediately affect the input/output behaviour in anomalous ways, e.g. a change in the gravity parameter of a quadcoptor's altitude control software ($UAVSimG$), and c) **untested usage scenarios** manifested as external inputs to the system, which may not have a deviant measurement distribution parameter, e.g. an electromagnetic attack on a sensor decreasing its fidelity ($UAVEMA$) or a phantom meal, where a user of a insulin delivery ADD announces a meal without ingesting any to trick it for a high insulin dose. As such, U2s may not result in an out-of-distribution (OOD) input or output, rather in an OOD inter-relationship among measured variables, necessciating a *multi-variate time series analysis* for the U2 detection.

Multi-variate time series anomaly detection (MTAD) is of recent research interest with a plethora of techniques ranging from statistical regression methods e.g ARIMA (Schmidt et al., 2018), Kalman filter (Huang et al., 2023), principal component analysis based techniques (Shyu et al., 2003), methods that use autoencoders (Borghesi et al., 2019), long short term memory (LSTM) based deep learning (DL) techniques, transformers (Tuli et al., 2022) and most recently large language models (LLMs) (Alnegheimish et al., 2024). The general technique (Figure 1 Panel A) has three steps: a) **training**: that creates a high dimensional latent space representation of the normal operation using data that may or may not have anolmalies but do not have anomaly labels, b) **validation**, that uses data with anomalies but without anomaly labels to learn a *anomaly score threshold* such that two fairly separated clusters are found in the validation set using the peaks over threshold method guided by the extreme value theory (Siffer et al., 2017), and c) **evaluation**, where anomaly score of successive overlapping / non-overlapping windows of test data are computed and compared with

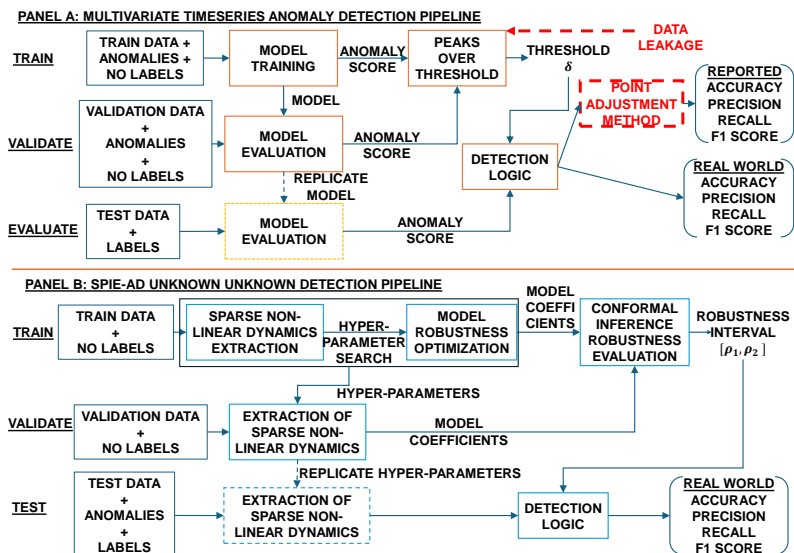

Figure 1: Panel A: SOTA MTAD pipeline with the identified issues highlighted by dashed arrows and boxes. Panel B: SPIE-AD's approach for solving zero-shot MTAD problem.

the threshold to determine anomalous data. There are three major problems with the state-of-the-art MTAD approach results in **unrealistic performance on benchmark datasets**:

**a) A1: Use of data leakage to learn anomaly score threshold -** In state-of-the-art (SOTA) MTAD techniques the validation set is same as the test data (refer to line 196 to 200 in the `data_loader.py` code in `https://github.com/thuml/Anomaly-Transformer`). This leads to potential data leakage and overfitting of the model. It is standard machine learning practice to keep validation set separate from test data. By definition, no validation dataset with anomalies are available for U2 or ZSA detection.

**Snippet of Results** (AT – Anomaly Transformer, GNAF – Graph Augmented Normalizing Flows)

| SPIE-AD evaluation on benchmarks SMD, SMAP, MSL [Xu et al. ICLR'22] | Average F1 scores (F1) and precision (P) across three benchmark datasets for MTAD (Exhaustive metrics in Results Section) | | | |
| --- | --- | --- | --- | --- |
| | Validation with data leak | | Validation without data leak | |
| | With point adjustment | Without point adjustment | With point adjustment | Without point adjustment |
| AT [ICLR'22] | F1:90 ±2,P:98 ±3 | F1:25 ±13,P:9±6 | F1:0±0,P:0±0 | F1:0 ±0,P:0±0 |
| GNAF [ICLR'22] | F1:74±4,P:75 ± 8 | F1:33 ±9,P:38±8 | F1:1.5 ± 2, P: 3 ± 2 | F1:0.1±0,P:0.1±0 |
| AnomalySimpleton | *F1:92±4, P:91±6* | F1:4±1,P:23±10 | F1:0±0, P:0±0 | F1:0±0,P: 0±0 |
| SPIE-AD + SINDY* | Not applicable | Not applicable | **F1: 78±12, P: 83±7** | **F1:77±9, P:81±6** |
| SPIE-AD + LTCNN* | Not applicable | Not applicable | **F1:84±11,P:85±9** | F1: 82±4,P: 85±9 |

Figure 2: Snippet of SPIE-AD performance for zero-shot MTAD against recent MTAD works on benchmark datasets.

**Technical difficulty in ZSA detection violating A1:** To the best of our knowledge, there is only one solution for zero-shot MTAD (Audibert et al., 2020). However, as identified by (Kim et al., 2022), it has poor realistic performance. Solutions for univariate zero-shot anomaly detection including techniques with LLMs (Alnegheimish et al., 2024) are available which as admitted by the authors are very difficult to adapt to MTAD. The technical challenge is to detect anomalies with no knowledge about anomalous data distribution, which preempts any discriminative feature learning methods.

**b) A2: Unrealistic evaluation method-** According to (Kim et al., 2022; Wu & Keogh, 2023), the reported results in nearly all state-of-the-art MTAD techniques have point adjustment (PA) (Su et al., 2019). This technique assumes that anomalies occur in contiguous segments, and if the MTAD method detects one point in this segment as anomalous, then every point in that segment should be considered as anomalous even if the MTAD method marks them as normal. The PA method inflates the precision by a significant amount Wu & Keogh (2023) in nearly all MTAD methods as seen in Figure 2), which shows the implementation of two most recent MTAD technique on benchmark datasets (SMAP, SMD, MSL discussed in more detail in Evaluation section) with code available from (Liu et al., 2024). These results are also supported by (Kim et al., 2022), which proposed an alternate evaluation criteria $PA\%K$, where PA is only employed if the original technique identifies $K\%$ of time points in an anomaly segment as anomaly. $K = 0$ indicates application of PA in its original form, while $K = 100$ indicates no PA.

**Technical difficulty in ZSA detection violating A2:** As highlighted in (Kim et al., 2022), in many real-world datasets, anomaly injection and manual labelling may result in several anomaly data-points to have similar distribution as normal data. So, if a MTAD method focuses only on latent features of data, its at inherent disadvantage in detecting anomalies.

**c) A3: Sensor data distribution shift due to anomaly:** U2 is a special case of anomaly, where there may not be a difference in the distribution parameters of the sensor outputs. Consider the example U2 scenario of wrongful Maneuvering characteristics augmentation system (MCAS) trigger in the fateful flight of Lion Air (Curran et al., 2024). MCAS was designed to mask the flight characteristics changes that would have occurred on newer Boeing Max 8 aircrafts (Herkert et al., 2020). This implies that if MCAS is wrongfully triggered then by design it attempts to make the distribution parameters of the flight characteristics similar to a normal flight. Figure 3 shows the data distribution of all sensors for anomalies and normal data in benchmark MTAD datasets in Panel A and for U2 and normal scenarios in Panel B. The Kolmogorov-Smirnov (KS) hypothesis test (KS, 2008) is used to compute the normalized maximum difference in cumulative distribution function (CDF) between normal and anomalous/U2 data (H = 1 implies the two distributions are statistically different with $(1-P)$ probability. Higher value of the CDF difference implies more deviant distribution). It's seen while in benchmark datasets anomalous and normal data have significantly different distributions, in our U2 datasets, distribution differences between U2 and normal data are insignificant.

**Technical difficulty in ZSA detection violating A3:** A3's violation implies the raw sensor data may not have latent information to discriminate between normal and U2 classes. So, any data-driven feature based method e.g. existing MTAD methods may not be useful. While the sensor data distributions may not be discriminative, there maybe a change in functional relationship among the sensors. Panel C shows the underlying nonlinear dynamical model mined from U2 and normal data using SINDY-MPC (Kaiser et al., 2018) has significantly different distribution parameters. ZSA detection could utilize modeling and monitoring of variations in such inter-relationships.

**Main Technical Contribution:** *We present* **SPIE-AD**, *that detects U2 by solving the general problem of zero-shot MTAD while violating the assumptions A1, A2 and A3 of SOTA MTAD methods.* The backbone of **SPIE-AD** are the *two fundamental theoretical contributions* of this paper: a) **robust sparse non-linear dynamical model recovery** from real-world multi-variate data using neural architectures with automated differentiation (AD) and b) **statistical conformance based model robustness interval extraction (CRIE)** method that can identify statistically relevant difference in recovered models. Utilizing these, *SPIE-AD* implements the following ZSA detection pipeline (Pane B in Figure 1): a) **training phase:** where **SPIE-AD** mines several models from training data snippets and determines a model robust-

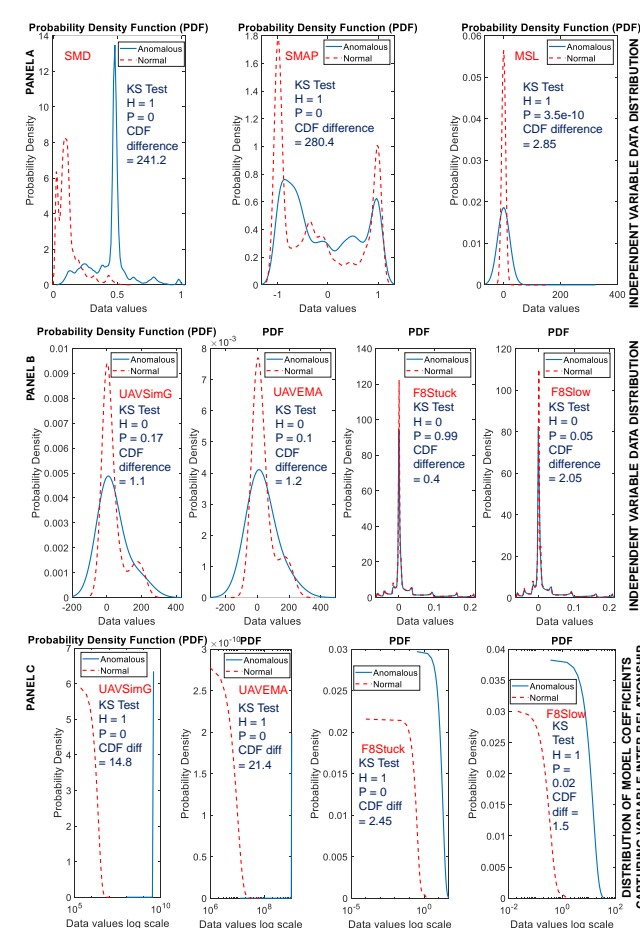

Figure 3: Panel A: Normal versus anomalous data distribution difference in benchmark datasets for evaluating MTAD methods, Panel B: U2 datasets have negligible distribution difference with normal. Panel C: significant distribution difference in parameters of U2 versus normal data in the underlying sparse model space.

ness metric using difference between two models quantified with standard distance measures, b) **validation phase:** it uses part of the training data in the CRIE algorithm to determine a robustness interval, and c) **evaluation phase:** it continually mines models from test data, computes robustness and compares with robustness interval to determine anomalies.

**Benchmark Contribution:** We introduce six synthetic benchmarks derived from commonly occurring U2 scenarios in three different types of real-world systems including quadcoptor, F8 cruiser,

and automated insulin delivery (AID). The hallmark of these benchmarks is that there is statistically insignificant distribution shift between the anomalous and normal data in each time series.

**Evaluation Contribution:** We first show that if we use point adjustment (K = 0) and allow for data leakage to obtain the optimal threshold for anomaly score, then it is possible to develop an untrained simpleton machine (AnomalySimpleton in Figure 2) that can beat state of art MTAD techniques. While this was also argued in (Kim et al., 2022), we propose a deterministic algorithm that gives consistent performance across the benchmark datasets used in baseline MTAD techniques. We evaluate recently proposed MTAD techniques along with **SPIE-AD** under realistic scenarios where the precision is not augmented with PA (i.e. K = 100) and anomaly signatures in the form of validation set is not available for threshold learning. All code and datasets available in supplement.

## 2 METHODOLOGY AND THEORETICAL FOUNDATIONS

**Problem Definition:** We consider $n$ sensors each with time series $X^i$ for sensor $i$ forming a vector $X(t)$ over time where $t \in 0 \ldots N/\mu$, where $\mu$ is the sampling frequency. The dataset consists of three sets: a) training set $Xtrain$, where no anomaly labels are available, b) $Xtest$, where there is a mix of anomalous and normal data and a corresponding label set $y(t)$, where $y(t) = 1$ if the time point $t$ in the test data is anomalous or $y(t) = 0$ if normal. The **zero-shot anomaly detection** problem is to use $Xtrain$ to learn a machine that can provide $\tilde{y}(t)$ which is an accurately estimate of $y(t)$ for the test set $Xtest$ without using any part of the test set $Xtest$ during model training.

**Method:** The main hypothesis of **SPIE-AD** is that input / output time-series data from autonomous systems must satisfy physical/chemical/mechanical/physiological properties of the real world system. Such properties are typically expressed using sparse non-linear dynamical systems:

$$\dot{X}(t) = f(X(t), \omega, t), \tag{1}$$

where $X(t)$ is the multivariate timeseries of dimension $n \times 1$, $n$ is the total number of variables, available at $N$ number of time steps at sampling frequency $\mu$, $\omega$ is the set of $p$ model coefficients that defines the sparse model. An $n$-dimensional model with $M^{th}$ order non-linearity can utilize $\binom{M+n}{n}$ non-linear terms. A sparse model only includes a few non-linear terms $p << \binom{M+n}{n}$.

### 2.1 ROBUST SPARSE DYNAMICAL MODEL RECOVERY

Given $N$ time sequenced measurement of $X(t)$, sparse model recovery (SMR) aims to recover the coefficient $\omega$ such that the reconstructed measurements $Y(t)$ by solving the ordinary differential equation (ODE) in Equation 1 satisfies an error threshold $\epsilon$, i.e., $\sum_{t=1}^{N} ||Y(t) - X(t)||^2 < \epsilon$.

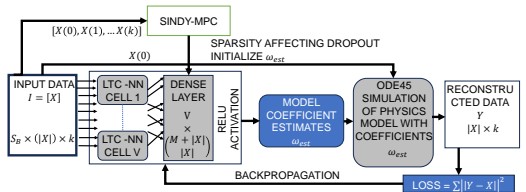

SMR is a well-researched problem with solutions ranging from L2 minimization techniques with sparse regression (SINDY-MPC) (Kaiser et al., 2018) to physics informed neural networks (PINN) (Chen et al., 2021). It is generally acknowledged that SOTA MR techniques suffer significant performance degradation on data from real world systems (O'Brien et al., 2023). This implies that with low sampling frequency and high noise (low signal to noise ratio), the model coefficients $\omega_i$ and $\omega_j$ derived from two consecutive segments $[i, i+k]$, and $[j, j+k]$ of $X(t)$, with window size $k$ has significant variance. This is problematic for **SPIE-AD** since it will be difficult to distinguish between noise and real U2 scenarios and will hamper the false positives. **SPIE-AD** needs model recovery that is robust to measurement noise under low sampling rates.

Figure 4: Robust model recovery technique where SINDY-MPC is used to initialize the model coefficients and the sparsity is used to configure the dense layer of the learning network. The LTC-NN architecture then refines the model coefficients by optimizing model recovery error under measurement noise and preventing model divergence.

To address robustness, **SPIE-AD** integrates SINDY-MPC with neural networks with automated differentiation, specifically liquid time constant neural networks (LTC-NN) as shown in Figure 4. Given a segment with $k$ samples, the SINDY-MPC technique is used to first recover a sparse model coefficient estimate $\omega(0)$. The same data segment is passed through a fully connected network of $V$ LTC-NN cells. This is done in batches of $S_B$. The output of the LTC-NN nodes are then fed to a dense linear layer with $\binom{M+n}{n}$ nodes with RELU activation function. The sparsity of $\omega(0)$, i.e. which elements of $\binom{M+n}{n}$ is "0" is used to dropout nodes of the dense layer and the value of the elements of $\omega(0)$ is used to constrain the dense layer outputs. A simple threshold based technique is used where the output of the $i^{th}$ dense layer node can only range between $[(1-\psi)\omega_i, (1+\psi)\omega_i]$, $\psi$ is a hyperparameter. The weighted dense layer output is the refined estimate $\omega_{est}$ of the model coefficients

and is fed to an ODE45 solver (Shampine et al., 2003) that reconstructs the signal $Y$. The loss is the mean square error between $X$ and $Y$ summed over dimensions and time steps. Here, we show the direct effect of using this robust model recovery method on ZSA detection. In supplementary document Table S1, we specifically evaluate the robustness of the models recovered by LTC-NN approach on standard SMR benchmarks in (Kaheman et al., 2020; Kaiser et al., 2018).

## 2.2 CONFORMAL INFERENCE FOR MODEL DEVIATION

Conformal inference (Krichen & Tripakis, 2004) is used to identify whether a new model generated from a window $[i, i + k]$ from validation data $\omega^v$ is in the distribution of the set of models learned during training $\Omega$ measured using a robustness metric $\rho$ in Equation 2.

$$\rho(\omega^v, \Omega) = (\sum_{i=1}^{|\Omega|} \Omega_i^T \omega^v)/|\Omega|, \tag{2}$$

where $|\Omega|$ is the number of elements in the set $\Omega$ and $\Omega_i^T$ denotes transpose of an element in $\Omega$.

Let us consider that the training data has $W$ windows of size $k$ each, $X_1(1 \ldots k), X_2(1 \ldots k), \ldots X_W(1 \ldots k)$. Also lets assume that each window is i.i.d in $\mathcal{R}^n \times \mathcal{R}^k$ drawn from a distribution $\mathcal{D}_X$. The SMR mechanism $L$ is used to derive coefficients $\omega_i \in \mathcal{R}^p$ from each $X_i$ such that reconstruction error is less than $\epsilon$. We use the same $L(.,.)$ to derive $\omega_{m+1}^v$ for $X_{m+1}, Y_{m+1}$ in validation data with no assumption on the $\mathcal{D}_{XY}$, hence no anomaly is required in validation set. Given the robustness function $\rho(.,.)$ in Equation 2, conformal inference creates a prediction band $C \subset \mathcal{R}^2$ based on $(X_1, Y_1), (X_2, Y_2), \ldots (X_m, Y_m)$ for a given $\alpha \in \{0, 1\}$, also called the *miscoverage level*, such that

$$P(\rho(\omega_{m+1}^v) \in C) \geq 1 - \alpha, \tag{3}$$

Split conformal prediction (Tibshirani et al., 2019) was proposed to construct prediction intervals that satisfy properties such as Eqn. 3. The prediction process can be encoded in Algorithm 1 CRIE, which takes the i.i.d training data $(X_1, Y_1) \ldots (X_m, Y_m)$, miscoverage level $\alpha$ and the SMR method $L$ to provide the prediction interval. The basic method is to divide the training set into two mutually exclusive subsets $I_T$ for training and $I_V$ for validation. The SMR method $L$ is used to derive $\omega_i$ for the segments $(X_i, Y_i) \in I_T$ and form the set $\Omega$. For each $\omega_i \in \Omega$, $\rho(\omega_i, \Omega_{/\omega_i})$ is computed, where $\Omega_{/\omega_i}$ denotes the set $\Omega$ with $\omega_i$ removed. Let $\sigma = avg_i(\rho(\omega_i, \Omega_{/\omega_i}))$ be the mean value of the robustness metric in the training set.

From the validation set, $\omega_j^v$ is derived for $(X_j, Y_j) \in I_V$. The residual $\rho(\omega_j^v, \Omega) - \sigma$ is derived for every element in $I_V$, the residual is arranged in ascending order. The algorithm then finds the residual at the position $\lceil (|I_V|/2 + 1)(1 - \alpha) \rceil$. This residual is used as the prediction range $d$. Theorem 2.1 in Lei et al. (2018) proves that the prediction interval at a new point $(X_{m+1}, Y_{m+1})$ is given by $L$ and satisfies the Theorem 1.

---

**Algorithm 1 CRIE($\{X_i\}_{i=1}^N$,$\alpha$,$\rho(.,.)$,$L$)**

1: **input** Data $\{X_i\}_{i=1}^N$, miscoverage level $\alpha$, robustness function $\rho$, SMR function $L$
2: **output** Confidence range $d$
3: Split $\{1, \ldots, N\}$ into two equal sized subsets $I_T$ and $I_V$.
4: $\omega_i = L((X_i) : i \in I_T)$
5: $\omega_j^v = L((X_j) : j \in I_V)$
6: Average robustness $\sigma = avg(\rho(\omega_i, \Omega_{/\omega_i}))$
7: For each $\omega_j^v$ compute residual $R_j = \rho(\omega_j^v, \Omega) - \sigma$
8: **return** $d$ = the kth smallest value in $\{R_j : j \in I_V\}$, where $k = \lceil (|I_V|/2 + 1)(1 - \alpha) \rceil$

---

**Theorem 1** *If $\Omega$ is a set of coefficients s.t. $L(X_i, \omega_i)$ and $X_i$ satisfy error margin $\epsilon$, then for a new $\omega_{m+1}^v$, $(X_{m+1}, Y_{m+1})$ and a $d$ from Algorithm 1, $P(\rho(\omega_{m+1}^v, \Omega) \in [\sigma - d, \sigma + d]) \geq 1 - \alpha$.*

## 2.3 ZSA DETECTION ALGORITHM

Utilizing Theorem 1 and the CRIE algorithm, we derived a robustness range depending on the robustness metric that encodes the normal behavior of the autonomous system if training data is U2 free. Our ZSA detection mechanism in Algorithm 2 simply takes windows of test data, uses the SMR technique to learn the model coefficients $\omega_i$, computes the robustness using Equation 2, computes residual, and compares with the range given in Theorem 1.

## 2.4 WHY THIS WORKS?

**How SPIE-AD addresses A1?** The robust model learning mechanism captures variable inter-relationships rather than individual sensor data characteristics. The **CRIE** algorithm then learns a tight range within which the robustness evaluation of the inter-relationship should fall for normal operation. Thus any deviation of inter-relationship beyond this range can be categorized as U2. Hence, **SPIE-AD** does not need a validation set with anomalies.

**How SPIE-AD addresses A2?** Unlike SOTA MTAD, **SPIE-AD** extracts low dimensional representation of the data which essentially reduces entropy, making it easier to model normal scenarios. U2 scenario lead to exaggerated model deviation since the inter-relationship between variables become inconsistent. Hence, as seen in Table 3, **SPIE-AD** can achieve better overall precision without PA.

**How SPIE-AD addresses A3?** By learning an underlying model, SPIE-AD can exploit significant distribution differences in model space of U2 scenarios (Figure 3).

---

**Algorithm 2 ZSADetect**$(\{X_i\}_{i=1}^{W}, \rho(.,.), L, \sigma, d, \Omega)$

1: **input** Test data $\{X_i\}_{i=1}^{W}$ with U2, robustness function $\rho$, SMR function $L$, mean robustness $\sigma$, interval $d$ from CRIE algorithm, and $\Omega$ set of all coefficients recoverd from training set.
2: **output** U2 label
3: $\omega_i = L((X_i) : i \in 1 \ldots W)$
4: Compute residual $R_i = \rho(\omega_i, \Omega) - \sigma$
5: **if** $R_i \in [\sigma - d, \sigma + d]$ **then**
6:     mark all samples in the window $X_i$ as 0 (not U2)
7: **else**
8:     mark all samples in the window $X_i$ as 1 (U2)
9: **end if**
10: **return** U2 labels

---

## 2.5 COMPUTATIONAL COMPLEXITY

There are two model recovery cores of SPIE-AD: SINDY-MPC and LTC-NN. SINDY-MPC uses the sequential threshold ridge regression (STRidge) (Kaiser et al., 2018) strategy. The computational complexity of Ridge regression in the worst case is $O(Nn^2)$, where $N$ is the number of samples and $n$ is the dimension of the multivariate signal if the number of regularization parameters is less than $N$ (Wang & Pilanci, 2023), which is the case in the example of anomaly detection. The sequential threshold runs Ridge regression multiple times until a desired reconstruction accuracy is obtained. If we fix a maximum $Q$ number of times that the sequential threshold can run then the overall computational complexity of SINDY-MPC is $O(QNn^2)$.

For the LTC-NN architecture, the computation complexity of forward pass is $O(V + V(|\Theta| + q)) + O(|X|N)$, where $N$ is the number of samples in the data, $V$, $q$, $\Theta$, $X$ are as in Figure 4. Complexity of backward pass is $O(VP_{LTC}N + V(|\Theta| + q)P_{dense}N)$, where $P_{LTC}$ is the number of parameters in the LTC cell, and $P_{dense}$ is the number of parameters in each neuron of the dense layer. SINDY-MPC on a single CPU thread was 11.3 ($\pm$ 2.1) times faster than the neural architecture on GPU. The overall computational complexity is $O((N/W)QNn^2)$ for SPIE-ADS and $O((N/W)VP_{LTC}N + V(|\Theta| + q)P_{dense}N)$, where $W$ is the window size of CRIE.

Table 1: Related works in MTAD. Italicized text are the baselines.

| Works | MTAD | Zero shot | Violates A1 | Violates A2 | Violates A3 |
|---|---|---|---|---|---|
| Pure statistical approaches | | | | | |
| Extended Kalman Filter (Huang et al., 2023) | No | Yes | Yes | Yes | No |
| Principle Component Analysis (Shyu et al., 2003) | Yes | No | No | No | No |
| Time series analysis methods | | | | | |
| *Time frequency anomaly detection (Zhang et al., 2022)* | Yes | No | No | No | No |
| *Frequency Interpolation Time Series (Xu et al., 2024)* | Yes | No | No | No | No |
| Statistical Machine Learning approaches | | | | | |
| K nearest neighbor (Wang et al., 2020) | Yes | No | No | No | No |
| *Isolation Forest (Liu et al., 2008)* | Yes | No | No | No | No |
| *Light weight online anomaly detection (Pevný, 2016)* | Yes | No | No | No | No |
| Deep learning models | | | | | |
| *OmniANomaly (Su et al., 2019)* | Yes | No | No | No | No |
| *Anomaly transformers (Xu et al., 2022)* | Yes | No | No | No | No |
| *Graph attention networks (Zhou et al., 2020)* | Yes | No | No | No | No |
| *LSTM (Hundman et al., 2018)* | Yes | No | No | No | No |
| *Graph augmented normalized flows (Zhao et al., 2022)* | Yes | No | No | No | No |
| *One size fits all (Zhou et al., 2023)* | Yes | No | No | No | No |
| Zero shot MTAD approaches | | | | | |
| *Usupervised anomaly detection (Audibert et al., 2020)* | Yes | Yes | Yes | No | No |
| CLIP zero shot image recognition (Pratt et al., 2023) | No | Yes | Yes | Yes | No |
| LLM Anomaly detection (Alnegheimish et al., 2024) | No | Yes | Yes | Yes | No |
| **SPIE-AD** | **Yes** | **Yes** | **Yes** | **Yes** | **Yes** |

## 3 RELATED WORK

Development of anomaly detection techniques (Table 1) has a rich history starting from univariate anomaly detection in time series with initial works employing Kalman Filter (Huang et al., 2023) and principle component analysis (PCA) (Shyu et al., 2003). While Extended Kalman Filter based techniques have been proposed for mode identification with multi-variate data (de Bézenac et al., 2020), they have not been used for MTAD. On the other hand, PCA has been used for MTAD but not zero shot. The next generation MTAD techniques used statistical learning methods such as K nearest neighbors (Wang et al., 2020) or Isolation Forest (iForest) (Liu et al., 2008) mechanisms or light weight online anomaly detector (LODA) (Pevný, 2016). Such techniques are not tested for zero shot MTAD and also had poorer overall performance on real world data (Liu et al., 2024). Recent works have also utilized time series analysis methods such as time frequency domain approaches (Zhang

Table 2: Benchmark datasets.

| Dataset | Dim | Total samples (Train/Test) | U2 / Anomaly % | Real world / Synthetic |
|---|---|---|---|---|
| UAV electromagnetic attack ($UAVEMA$) | 3 | 240K/242K | 29.75% | Synthetic |
| UAV simulated g change ($UAVSimG$) | 3 | 240K / 274 K | 11.7% | Synthetic |
| F8 cruiser stuck elevator ($F8Stuck$) | 4 | 877K/237K | 9.2% | Synthetic |
| F8 cruiser slow elevator ($F8Slow$) | 4 | 877K / 843 K | 1.4% | Synthetic |
| AID phantom meal ($AIDPhantom$) | 4 | 260K/240K | 12% | Synthetic |
| AID cartridge error ($AIDCartridge$) | 4 | 260K / 302 K | 11.5% | Synthetic |
| Server Machine Dataset (SMD) | 38 | 708K / 708K | 4.16% | Real world |
| Soil Moisture Active Passive Satellite (SMAP) | 25 | 135 K / 427 K | 13.13% | Real World |
| Mars Science Lab Rover (MSL) | 55 | 58 K / 73 K | 10.7% | Real World |
| UCR anomaly detection dataset | 1 | 5302K / 13846K | 0.4% | 250 Real World |

et al., 2022) or frequency interpolation methods (Xu et al., 2024) to perform MTAD. The current generation of MTAD techniques uses DL and include use of LSTM (Hundman et al., 2018), variational autoencoders (OmniAnomaly) (Su et al., 2019), anomaly transformers (AT) (Xu et al., 2022), graph augmented normalized flows (GNAF) (Zhao et al., 2022), and Graph Attention Networks (GAT) (Zhou et al., 2020) or even language model based one size fits all (OFA) approach (Zhou et al., 2023). These MTAD techniques however use the workflow described in Figure 1 and do not achieve zero shot MTAD. While zero shot anomaly detection has been explored in the image domain using large vision models such as CLIP (Pratt et al., 2023) such methods are not directly applicable to zero shot MTAD. We are aware of two works, i) unsupervised anomaly detection (USAD) that performs zero shot MTAD (Audibert et al., 2020) using autoencoders, and ii) and one that uses large language models (LLMs) to perform zero shot anomaly detection in univariate timeseries (Alnegheimish et al., 2024). The USAD technique still reports anomaly detection accuracy with point adjustment and relies on difference in sensor data distribution between normal and anomalous class hence still does not violate $A2$ and $A3$.

## 4 EVALUATION

We perform three types of evaluation: a) effects of using test set as validation set (A1) and PA (A2) on anomaly detection performance. We show that an untrained statistical method can beat SOTA learning based systems with A1 and A2.

b) performance comparison of SPIE-AD and SOTA baselines under violation of A1 and A2 on U2 benchmarks that have no distribution shift between anomaly and normal data (violates A3).

c) performance comparison of SPIE-AD and SOTA baselines on real world univariate and multivariate datasets. We use the large univariate UCR dataset to perform a statistically robust evaluation of sensitivity of SPIE-AD on window size $W$.

**AnomalySimpleton:** We propose an untrained deterministic thresholding algorithm that exploits PA and test data distribution i.e. data leakage to provide anomaly detection performance on par with state-of-the-art learning techniques. In this method, a specific window $W$ of data is selected from the train data. Statistical properties of the train data window $W$ such as mean $\psi_{train}$, standard deviation $\sigma_{train}$, and skewness $\kappa_{train}$ is computed. For each test data window of length $W$, the same statistics are computed. If the deviation of the test statistics is more than P% of the train statistics, then the test data window is classified as anomalous else it is not anomalous. The window $W$ and the test statistics $P$ is used to obtain two maximally separated clusters in the test data. This is done through brute force search over several $W$ and $P$ options. For each benchmark real world data this window and threshold seach is performed from scratch.

### 4.1 BENCHMARKS

We used 9 datasets to evaluate **SPIE-AD**, out of which 6 are synthetic U2 dataset while 3 are real world anomaly datasets. U2 datasets are synthetic due to the rarity of real world U2 data and the associated confidentiality hurdles. While the synthetics datasets highlights the efficacy of **SPIE-AD** in ZSA detection while violating $A1$, $A2$, and $A3$, the real world anomaly datasets show the general applicability of **SPIE-AD** as a zero shot MTAD technique.

*F8 Cruiser:* This is an aircraft pitch control system using a model predictive control for trajectory tracking. The U2 scenario is a hardware failure where the elevator gets jammed and maintains a constant position despite the controller providing it varying inputs ($F8Stuck$). Another U2 scenario is the elevator responds slower than usual with low maximum angular velocity ($F8Slow$).

*UAV Altitude control:* This is a quadcoptor, whose altitude is controlled by four proportional integrative and derivative (PID) controllers. These controllers provide balanced thrusts in each propeller so that the UAV maintains a given height. The first U2 is a software attack that changes the gravity parameter $g$ in the controller software ($UAVSimG$). The second U2 scenario is an electromagnetic attack on the UAV gyroscope sensor ($UAVEMA$).

*Automated insulin delivery system:* This is an hybrid close loop autonomous system that decides on insulin delivery for an individual with Type 1 Diabetes. It works autonomously for the most part, but requires human intervention with extra insulin delivery to manage meal intake. One of the ways to trick the system to deliver a high dosage of insulin is to announce to the system that a large meal has been ingested without actually consuming the meal. This is called phantom meal and is the first U2 scenario in this domain ($AIDPhantom$). In the second scenario, the human participants poorly installs the insulin cartridge resulting in insulin occlusion or blockage. The block causes insulin build up since the AID system cannot monitor the cartridge error and finally it gives way and injects an overdose of insulin $AIDCartridge$.

Our U2 benchmarks cover the three categories of U2 scenarios discussed in the Introduction section. The $F8Stuck$, $F8Slow$, and $AIDCartridge$ are caused by hardware failure, the $UAVSimG$ and $UAVEMA$ are software failures, and the $AIDPhantom$ is an example of U2 arising from human interaction with autonomous systems.

In all the U2 examples, U2 scenarios are generated by selecting random times at which the U2 event is activated, with the duration of U2 activation also sampled from a random distribution.

**Real world datasets:** We use two types of real world databases: a) standard datasets available in (Su et al., 2019) and summarized in Table 2, and b) UCR database, a large set of 250 real world anomaly datasets available in (Wu & Keogh, 2022). Detailed dataset description is in supplement.

**Baseline Techniques:** We compare **SPIE-AD** with several deep learning based techniques that follow the well established pipeline for anomaly detection as introduced in (Su et al., 2019). In addition, we also compare our technique to the only other zero shot MTAD approach available in recent literature. All baseline techniques are highlighted in italics in Table 1.

### 4.2 IMPLEMENTATION

**SPIE-AD implementation:** We implemented two variations of **SPIE-AD**: a) **SPIE-ADS**, where the model recovery part is solely SINDY-MPC, and b) **SPIE-ADL**, where the model recovery part is SINDY-MPC augmented with the LTC-NN neural architecture with AD. For the SINDY-MPC implementation we used the code from (Kaiser et al., 2018). For the LTC-NN neural architecture, we updated the base code available in (Hasani, 2024). The **CRIE** and **ZSA detection** algorithms were developed in house using Matlab 2022b. All code is available in supplementary document.

*Hyper-parameter optimization:* As highlighted in Figure 1, there is a hyper-parameter optimization step in **SPIE-AD** during the training process. The hyper-parameters include: a) miscoverage level $\alpha$ that determines the robustness interval width $d$, b) the polynomial order of SMR technique, c) the sparsity level of the model, and the window size $k$. These parameters were determined only using the training data with the objective to include atleast $r > 80\%$ points of the training dataset within the robustness interval while minimizing $d$. The hyper-parameter optimization approach was brute-force and performed for each application, but remained same for different U2s.

**Baseline Implementation:** We used the MTAD tools and pipeline established in (Liu et al., 2024) for baseline implementations. In all baseline implementations except USAD, we observed that removing labels from validation set reduced the precision and recall to near zero. Indicating that a pure zero-shot MTAD implementation with baselines is not possible without significantly altering the methods. Hence, in our comparison all baselines were non zero-shot MTAD except for USAD and **SPIE-AD**. For all implemented techniques we show two cases with and without PA.

**Evaluation metrics:** We use standard metrics: Precision (Pr), Recall (Re), and F1 score (Liu et al., 2024). For the univariate real-world UCR database, the event-based AD accuracy is used as in Timeseriesbench (Si et al., 2024). If the detected anomaly sample is in $\pm 100$ samples of the anomaly start point, accuracy is 1, else 0. Plus we show execution times of all methods for real world datasets.

## 5 RESULTS

We first show the inefficacy of the evaluation strategy used in state of the art MTAD techniques. We then evaluate the performance of **SPIE-AD** and compare with baseline on U2 benchmarks. We then compare **SPIE-AD** performance on real datasets. Here we also perform two ablation studies: a) removing point adjustment, and b) removing acess to validation datasets with anomalies.

### 5.1 ANOMALYSIMPLETON PERFORMANCE AND LESSONS LEARNED

Table 4 shows AnomalySimpleton could utilize PA and data leakage to beat GANF (Zhao et al., 2022) and USAD (Audibert et al., 2020) baselines on all real benchmark datasets and was on par with Anomaly Transformers (Xu et al., 2022). However, when PA was eliminated, its F1 score drastically dropped. Moreover, if data leakage was disabled, then its F1 score became 0. This shows a worse

Table 3: Comparison of **SPIE-AD** against baseline techniques for U2 benchmark examples. **SPIE-ADS** uses SINDY-MPC for SMR, while **SPIE-ADL** uses the LTC-NN architecture for SMR. $^+$ denotes with point adjustment (PA) and absence of $^+$ is without PA.

| Approach | $F8Stuck$ | | | $F8Slow$ | | | $UAVSimG$ | | | $UAVEMA$ | | | $AIDPhantom$ | | | $AIDCartridge$ | | |
|---|---|---|---|---|---|---|---|---|---|---|---|---|---|---|---|---|---|---|
| | Pr | Re | F1 | Pr | Re | F1 | Pr | Re | F1 | Pr | Re | F1 | Pr | Re | F1 | Pr | Re | F1 |
| OmniAnomaly$^+$ | 91.2 | 72.7 | 80.9 | 88.4 | 71.1 | 78.8 | 92 | 77.1 | 83.9 | 90 | 67.3 | 77.0 | 94 | 76.1 | 84.1 | 97 | 59.7 | 74 |
| OmniAnomaly | 41 | 26.8 | 32.4 | 65 | 28.1 | 39.2 | 32 | 19.7 | 24.4 | 29 | 16.8 | 21.3 | 19.1 | 16.5 | 17.7 | 65 | 31.9 | 43 |
| AT$^+$ | 100 | 78.6 | 88 | 100 | 58.7 | 74.1 | 100 | 59.2 | 74.2 | 90 | 56.1 | 69.1 | 91 | 56.3 | 69.7 | 100 | 59.2 | 74 |
| AT | 85.5 | 75.8 | 80.3 | 34.2 | 32.8 | 33.5 | 35 | 33.5 | 34.2 | 33.9 | 32.4 | 33 | 34 | 32 | 33 | 34.3 | 33.8 | 34 |
| iForest$^+$ | 100 | 78.6 | 88 | 100 | 47.5 | 64.4 | 100 | 50.8 | 67.6 | 88.5 | 46.2 | 60.7 | 98.6 | 45.9 | 62.6 | 91.2 | 42.1 | 57.6 |
| iForest | 14 | 33 | 19.6 | 9.8 | 8.2 | 8.9 | 10.6 | 8.5 | 9.4 | 8.6 | 7.6 | 8.1 | 9.5 | 8.1 | 8.7 | 9.5 | 7.9 | 8.6 |
| LODA$^+$ | 100 | 72.6 | 84 | 100 | 20.7 | 34.3 | 96.9 | 18.5 | 31 | 88.5 | 14.9 | 25.5 | 95.8 | 16.8 | 28.6 | 99.2 | 17.2 | 29.4 |
| LODA | 88 | 70 | 78 | 60.7 | 13.7 | 22.4 | 50.7 | 11 | 18 | 35 | 8.6 | 13.8 | 35.8 | 9.4 | 14.9 | 36.4 | 9.7 | 15.3 |
| LSTM$^+$ | 100 | 88 | 93 | 100 | 47.8 | 64.7 | 91.8 | 20.2 | 33.2 | 100 | 21.2 | 35 | 99.9 | 20.3 | 33.8 | 96 | 18.6 | 31 |
| LSTM | 77 | 85 | 80 | 61 | 35.8 | 45.2 | 59.4 | 13.2 | 21.6 | 60.8 | 14.2 | 23 | 58.6 | 12.6 | 20.7 | 54.7 | 12.1 | 19.9 |
| USAD$^+$ | 100 | 72.1 | 83.8 | 100 | 37.4 | | 92.6 | 21.8 | 35.3 | 90.3 | 21.6 | 34.9 | 94.6 | 25.2 | 39.8 | 97.1 | 28.6 | 44 |
| USAD | 81 | 67.7 | 74 | 55.3 | 14.2 | 22.6 | 51.2 | 12.3 | 19.8 | 49.2 | 12.1 | 19.4 | 52.6 | 12.1 | 19.7 | 58 | 8.8 | 15.2 |
| GANF$^+$ | 100 | 86 | 92.5 | 100 | 58 | 73 | 100 | 92.2 | 96 | 100 | 97 | 98.5 | 96.7 | 61.5 | 75 | 92.8 | 56.1 | 70 |
| GANF | 61 | 79 | 68.8 | 3.2 | 4.3 | 3.7 | 51.4 | 85 | 64.3 | 0.9 | 24.7 | 1.8 | 3.2 | 4.5 | 3.8 | 2.1 | 2.7 | 2.4 |
| GAT$^+$ | 100 | 85.2 | 92 | 100 | 47.2 | 64.1 | 99.2 | 48.3 | 65 | 86.4 | 44.6 | 58.8 | 92.8 | 48.1 | 63.4 | 99 | 49 | 65.6 |
| GAT | 71.4 | 80.5 | 75.7 | 58.9 | 34.5 | 43.5 | 59.2 | 32.3 | 41.8 | 50.4 | 28 | 36 | 54.5 | 28.9 | 37.8 | 57.2 | 30.3 | 39.7 |
| OFA$^+$ | 82.1 | 87.5 | 84.7 | 65.9 | 43.2 | 52.2 | 66.2 | 72.3 | 69.1 | 70.4 | 68 | 69.2 | 74.5 | 77.1 | 75.8 | 81.3 | 87.4 | 84.2 |
| OFA | 21.4 | 4.5 | 7.4 | 21.9 | 9.7 | 13.4 | 37.5 | 22.1 | 27.2 | 20.3 | 8.5 | 12 | 31.3 | 18.3 | 23.1 | 21.7 | 10.1 | 13.8 |
| FITS$^+$ | 91.4 | 70.5 | 79.6 | 81.3 | 74.2 | 77.6 | 81.9 | 82.3 | 82.1 | 80.1 | 76 | 78 | 74.3 | 88.1 | 80.6 | 97.2 | 70.1 | 81.5 |
| FITS | 21.4 | 8.6 | 12.3 | 48.1 | 14.3 | 22.05 | 17.3 | 21.9 | 19.3 | 80.4 | 2.4 | 4.7 | 24.5 | 18.4 | 21.0 | 14.7 | 40.1 | 21.5 |
| TFAD$^+$ | 82.1 | 77.4 | 79.7 | 78.2 | 84.3 | 81.1 | 91.9 | 82.3 | 86.8 | 80.4 | 88 | 84.0 | 71.5 | 78.9 | 75.0 | 87.2 | 80.3 | 83.6 |
| TFAD | 11.2 | 30.4 | 16.4 | 9.8 | 21.7 | 13.5 | 29.5 | 12.4 | 17.5 | 21.9 | 8.7 | 12.4 | 14.7 | 31.8 | 19.9 | 17.7 | 21.4 | 19.4 |
| **SPIE-ADS$^+$** | 87.3 | 100 | 93.2 | 54.8 | 100 | 71 | 82 | 100 | 90.1 | 91.1 | 100 | 95.4 | 94 | 98.1 | 96 | 95.3 | 93 | 94.1 |
| **SPIE-ADS** | 86.7 | 94.5 | **90.4** | 51 | 85 | **66** | 82 | 99.9 | **90.1** | 91.1 | 100 | **95.4** | 91 | 96 | **93.4** | 92 | 85 | **88.4** |
| **SPIE-ADL$^+$** | 88.9 | 100 | 94 | 55.1 | 100 | 73 | 91 | 100 | 95.3 | 93.2 | 100 | 96.5 | 94.1 | 99 | 96 | 95 | 94 | 94.1 |
| **SPIE-ADL** | 88.7 | 95.1 | **92** | 58 | 93 | **70** | 89 | 99.9 | **94.2** | 93.2 | 100 | **96.5** | 92.1 | 99 | **95.4** | 91 | 92 | **91.5** |

case machine with very poor realistic performance can result in a very good anomaly detection method through the usage of point adjustment and threshold learning using test data. Through this misadventure, we have learned the following lessons:

**Lesson 1:** anomaly detection works should show results for both with / without PA or use metrics such as $PA\%K$ as proposed in (Kim et al., 2022).

**Lesson 2:** anomaly detection works should explicitly address data leakage issue by either obtaining validation data from train set or ensuring that validation set and test set are mutually exclusive.

## 5.2 ZSA DETECTION PERFORMANCE EVALUATION

Table 3 shows that **SPIE-ADS** outperforms SOTA on the F1 score for the case without PA - implying it has better precision and recall and does not need PA. Methods such as anomaly transformers (AT) do outperform **SPIE-AD** in F1 metric with PA - implying **SPIE-AD** does miss some legitimate events as evidenced by the slightly higher recall. Interestingly, among the DL methods, AT has the highest difference between F1 scores with and without PA. However, AT has the highest F1 score for $F8Slow$. This entails that while anomaly trasnformer is good at detecting U2, albeit very late. Further, SPIE-AD also outperforms the only other zero-shot MTAD methods USAD. USAD also has a significant difference in metrics with/without PA (A2). SPIE-AD requires no such assumptions.

Another inference is that for nearly all cases **SPIE-ADL** consistently outperforms **SPIE-ADS**, showing the robustness improvement property of the LTC-NN approach in Figure 4. However, the difference is much lower and given that LTC-NN architecture is much more complex than SINDY-MPC, one may wonder why it's necessary. A point is that all these benchmarks are synthetic; hence are much less noisy reducing its need. However, the need for LTC-NN is illustrated in real data.

## 5.3 REAL WORLD ANOMALY DETECTION PERFORMANCE

**Multi-variate:** Table 4 shows the performance of SPIE-AD on real datasets and compares it to recent DL based MTADs and unsupervised methods. In real data, SPIE-AD outperforms SOTA on F1 score without PA. As expected on real data, we see the largest benefit of using the LTC-NN.

**Univariate:** Maximum event-wise AD accuracy of SPIE-ADS was $75.6\%$ on UCR (n = 250) database. Compared to the leaderboard in Lee et al. (2024), SPIE-ADS beats the SOTA by 4.8%.

**Ablation Studies:** For each real dataset we created three configurations: with point adjustment and validation set (PA + V), without PA ($\neg$ PA), and without validation set i.e. zero shot ($\neg$ V). It is observed that as expected the F1 score of SOTA DL techniques reduce drastically without PA. The USAD has lesser effect, while the SPIE-AD methods have the least effect of PA. Moreover, removal

Table 4: Comparison of SPIE-AD with latest baseline techniques on real world datasets satisfying A3 and ablation studies. Only F1 score is shown. Time denotes the execution time in minutes. AnomalySimpleton and SPIE-ADS were executed in Intel Core i7 10th Gen CPU. All others in NVIDIA RTX 6000 Ada engine with CUDA 12.5. All other metrics in supplement Table S2.

| Method | SMD | | | | SMAP | | | | MSL | | | |
|---|---|---|---|---|---|---|---|---|---|---|---|---|
| | A2 | ¬ A2 | ¬ A1 | Time | A2 | ¬ A2 | ¬ A1 | Time | A2 | ¬ A2 | ¬ A1 | Time |
| AT | 90.7 | 38.8 | 0 | 372 | 91.2 | 22.3 | 0 | 183 | 88.6 | 13.1 | 0 | 175 |
| GANF | 78.6 | 41.2 | 3.4 | 361 | 71.9 | 32.8 | 1.1 | 179 | 73 | 24 | 0 | 165 |
| USAD | 43.1 | 21.2 | 21.2 | 218 | 62 | 26 | 26 | 121 | 41 | 18 | 18 | 103 |
| OFA | 72.9 | 2.5 | 1.9 | 318 | 86.9 | 9.4 | 5.1 | 171 | 82.7 | 22.3 | 4.4 | 159 |
| FITS | 99.9 | 32.7 | 11.2 | 281 | 70.74 | 13.4 | 2.2 | 164 | 78.12 | 15.3 | 4.3 | 141 |
| TFAD | 89.3 | 21.7 | 4.1 | 211 | 96.3 | 35.4 | 7.7 | 135 | 96.4 | 40.1 | 8.8 | 122 |
| AnomalySimpleton | 96.2 | 2.0 | 0 | 21 | 90.5 | 4 | 0 | 7 | 89.5 | 4.8 | 0 | 6 |
| SPIE-ADS | 74 | 73 | 73 | 172 | 68 | 65 | 65 | 153 | 83 | 83 | 83 | 132 |
| SPIE-ADL | 86 | 86 | 86 | 323 | 79 | 73 | 73 | 208 | 83 | 83 | 83 | 178 |

of validation set reduces the F1 score to near zero for anomaly transformer and GNAF approaches showing that cannot be trivially extended for zero-shot MTAD. On the other hand, both USAD and SPIE-AD have higher F1 score for zero-shot MTAD, with SPIE-AD outperforming USAD.

**Sensitivity to window size:** We use the UCR database to evaluate sensitivity to window size for our approach as it has the largest number of real world datasets (n = 250) to ensure statistically stable results. The window size is varied as a percentage of the total dataset size for each database. Figure 5 shows that large window sizes reduces the accuracy of detecting an anomalous event since the event size maybe a small fraction of the window size. When the window size is too small, SINDY-MPC core fails to extract accurate models of the underlying governing dynamics - decreasing its accuracy. Hence, there is a optimal window size for each dataset.

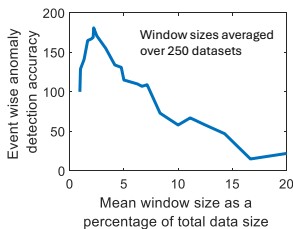

Figure 5: Event wise anomaly detection accuracy of SPIE-ADS with varying window size. Results averaged over 250 UCR datasets.

## 6 CONCLUSIONS AND DISCUSSION

In this paper, we introduced **SPIE-AD** a methodology for identifying 'unknown-unknown' (U2) errors in AI-enabled autonomous systems. U2 can arise due to unpredictable human interactions and complex real-world usage scenarios, potentially leading to critical safety incidents through unsafe shifts in the distribution of the inter-relationships among the variables in operational data. SPIE-AD performs zero shot anomaly detection and hence does not require signature of the U2 scenario or detection. Validation across diverse contexts such as zero-day vulnerabilities in unmanned aerial vehicles, hardware failures in autonomous insulin delivery systems, and design deficiencies in aircraft pitch control systems such as Maneuvering Characteristics Augmentation Systems (MCAS), demonstrates our framework's efficacy in preempting unsafe data distribution shifts due to unknown-unknowns. This methodology not only advances unknown-unknown error detection in AAS but also sets a new benchmark for integrating physics-guided models and machine learning to ensure system safety. Mining the underlying model of a dynamical system has several applications including detection of stealth cheating scenarios in AI systems much like the Volkswagon emission cheating case, or also biometric liveness detection.

We have not only shown efficacy of SPIE-AD on U2 datasets but also demonstrated its generality in detecting any anomalous scenarios through the usage of standard real world datasets. We will make our dataset public through the MTAD tools and techniques github page (Liu et al., 2024) for the general research community to develop novel ZSA detection schemes.

**Limitations:** SPIE-AD faces challenges in determining point anomalies that last very few samples. In the SMD SMAP and MSL datasets, anomalies that last $< 5$ samples are missed consistently. Moreover, as seen in Figure 5 SPIE-ADS performance is sensitive to the window size chosen for the CRIE algorithm. Hence, an important future work is to formally evaluate the sensitivity of SPIE-AD to window length.

**Ethical Considerations:** One of the components of SPIE-AD is recovering underlying model. One of the applications of SPIE-AD is digital twins. An unethical usage is impersonation. Thus, careful ethical evaluation is required when integrating such systems in medical practice. Another issue is that SPIE-AD is only a ZSA detection mechanism. In its current form it cannot be used to explain the reasons behind the U2 occurrence. Such black box models can become problematic if false positives lead to usage of critical intervention. Hence proper safeguards should be placed to vet the U2 decisions from SPIE-AD.

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

# A APPENDIX

## A.1 LTC-NN MODEL RECOVERY ROBUSTNESS RESULTS

Table S1 5 shows the performance of LTC-NN architecture described in Figure 4 of the main paper on model recovery for different benchmark examples available in (Kaiser et al., 2018).

For each evaluation experiment, we use two metrics:

**Root mean square error in model coefficients** ($RMSE_\Theta$) and **Root mean square error in signal** ($RMSE_Y$). Given the estimated model coefficients $\Theta_{est}$ and measured variables $Y_{est}$ for any technique we computed them as:

$$RMSE_\Theta = \sqrt{\frac{1}{p} \sum_{j=1\ldots p} (\Theta_{est}^j - \Theta^j)^2}, \tag{4}$$

$$RMSE_Y = \frac{1}{n} \sum_{l=1\ldots n} \sqrt{\frac{1}{k} \times \sum_{j=1\ldots k} (Y_{est}^l(j) - Y^l(j))^2}. \tag{5}$$

Table 5: S1: Comparison of LTC-NN architecture with baseline SINDY-MPC only and other RNN architectures on standard benchmarks. LTC-NN-MR represents model recovery with LTC-NN architecture shown in Figure 4. The LTC-NN can be replaced by CT-RNN or NODE. Value in () is standard deviation

| Example | RMSE | SINDY-MPC | LTC-NN-MR | CT-RNN-MR | NODE-MR |
|---|---|---|---|---|---|
| Lotka | $RMSE_\Theta$ | 0.059 (0.02) | 0.048 (0.015) | 0.054 (0.03) | 0.064 (0.02) |
| Volterra | $RMSE_Y$ | 0.03 (0.02) | 0.03 (0.018) | 0.05 (0.02) | 0.088 (0.03) |
| Chaotic | $RMSE_\Theta$ | 0.014 (0.008) | 0.015 (0.006) | 0.022 (0.009) | 0.044 (0.012) |
| Lorenz | $RMSE_Y$ | 1.7 (0.6) | 1.68 (0.4) | 3.66 (1.1) | 8.1 (3.6) |
| F8 | $RMSE_\Theta$ | 7.9 (3.2) | 6.8 (2.9) | 10.5 (4.8) | 19.9 (7.4) |
| Crusader | $RMSE_Y$ | 3.2 (2.1) | 1.57 (1.4) | 3.46 (2.6) | 7.22 (5.7) |
| Pathogenics | $RMSE_\Theta$ | 0.5 (0.2) | 0.39 (0.23) | 0.43 (0.3) | 0.42 (0.3) |
| attack | $RMSE_Y$ | 27.8 (9.1) | 28.3 (6.2) | 28.8 (7.7) | 29.5 (9.6) |

## A.2 DESCRIPTION OF REAL WORLD DATASETS

We used three real datasets:

**Server Machine Database:** The Server Machine Dataset (SMD) is a newly curated dataset that spans a period of five weeks, collected from a major Internet company known for its extensive server infrastructure (Su et al., 2019). This dataset, which includes detailed logs and metrics related to server machine performance, has been made publicly available on GitHub to support research in anomaly detection and related fields.

The SMD dataset comprises a wide range of features, including CPU utilization, memory usage, disk I/O, and network traffic, collected at regular intervals. For practical analysis, we have divided the dataset into two equal-sized subsets: the first subset, which covers the initial period of the data collection, is used as the training set. The second subset, covering the remaining period, is designated as the testing set.

In the testing subset, domain experts have meticulously identified and labeled anomalies, along with their specific dimensions, based on a thorough examination of incident reports and historical data. These labels provide valuable insights for evaluating anomaly detection algorithms and enhancing their accuracy.

**Soil Moisture Active Passive Satellite:** The Soil Moisture Active Passive (SMAP) satellite (Liu et al., 2024) is a NASA mission designed to measure and monitor soil moisture levels across the globe. SMAP employs a combination of active radar and passive radiometer technologies to provide high-resolution measurements of soil moisture, which are crucial for understanding water cycles, weather patterns, and climate change. The satellite records key performance indicators (KPIs) related to its operational status and performance metrics, including data on the satellite's health, instrument functionality, and environmental conditions. These KPIs are essential for ensuring the proper functioning of the spacecraft and for diagnosing and addressing any issues that may arise during its mission.

**Mars Science Laboratory Rover (MSL):** The Mars Science Laboratory (MSL) rover (Liu et al., 2024), commonly known as Curiosity, is a NASA rover mission designed to explore the surface of Mars. Equipped with a suite of scientific instruments, the MSL rover conducts a variety of experiments to study Mars' geology, climate, and potential for past habitability. The rover records KPIs

Table 6: S2: Comparison of SPIE-AD with latest baseline techniques on real world datasets and ablation studies. The datasets all satisfy A3.

| Method | SMD | | | | | | | | |
|---|---|---|---|---|---|---|---|---|---|
| | A2 | | | ¬ A2 | | | ¬ A1 | | |
| | Pr | Re | F1 | Pr | Re | F1 | Pr | Re | F1 |
| AT | 83 | 100 | 90.7 | 29 | 58.6 | 38.8 | 0 | 0 | 0 |
| GANF | 39.5 | 93 | 78.6 | 28 | 78 | 41.2 | 30.6 | 1.8 | 3.4 |
| USAD | 28 | 94 | 43.1 | 12.2 | 80 | 21.2 | 12.2 | 80 | 21.2 |
| AnomalySimpleton | 98.2 | 94.4 | 96.2 | 35.1 | 1.0 | 2.0 | 0 | 0 | 0 |
| SPIE-ADS | 64 | 87.7 | 74 | 63 | 86.7 | 73 | 63 | 86.7 | 73 |
| SPIE-ADL | 84 | 88 | 86 | 83 | 89 | 86 | 83 | 89 | 86 |
| **Method** | **SMAP** | | | | | | | | |
| | A2 | | | ¬ A2 | | | ¬ A1 | | |
| AT | 83.8 | 100 | 91.2 | 12.7 | 90 | 22.3 | 0 | 0 | 0 |
| GANF | 57.5 | 96 | 71.9 | 19.9 | 93 | 32.8 | 0.6 | 7 | 1.1 |
| USAD | 45 | 100 | 62 | 15.1 | 94 | 26 | 15.1 | 94 | 26 |
| AnomalySimpleton | 86.4 | 95.1 | 90.5 | 13.6 | 2.4 | 4 | 0 | 0 | 0 |
| SPIE-ADS | 55 | 89 | 68 | 52 | 87 | 65 | 52 | 87 | 65 |
| SPIE-ADL | 69.8 | 91 | 79 | 65.7 | 82.1 | 73 | 65.7 | 82.1 | 73 |
| **Method** | **MSL** | | | | | | | | |
| | A2 | | | ¬ A2 | | | ¬ A1 | | |
| AT | 79.5 | 100 | 88.6 | 8.7 | 27 | 13.1 | 0 | 0 | 0 |
| GANF | 64 | 85 | 73 | 16 | 48 | 24 | 0 | 0 | 0 |
| USAD | 44.5 | 38 | 41 | 14.5 | 23.8 | 18 | 14.5 | 23.8 | 18 |
| AnomalySimpleton | 89.6 | 89.4 | 89.5 | 20.9 | 2.7 | 4.8 | 0 | 0 | 0 |
| SPIE-ADS | 80.2 | 86 | 83 | 80.2 | 86 | 83 | 80.2 | 86 | 83 |
| SPIE-ADL | 80.3 | 85.8 | 83 | 80.3 | 85.8 | 83 | 80.3 | 85.8 | 83 |

related to its operational performance, such as power consumption, temperature readings, and communication status. These performance metrics are critical for monitoring the health and functionality of the rover, managing its systems, and troubleshooting any technical challenges that arise during its exploration of the Martian surface. The data collected helps scientists and engineers ensure the rover's effective operation and mission success.

## A.3 EXTENDED TABLE FOR REAL WORLD DATASET

Table S2 6 shows the extended results for Table 4 in the main paper with precision and recall values.

## A.4 SPIE-AD HYPER-PARAMETER OPTIMIZATION

Given a threshold of $r\%$, the hyper parameters of the SPIE-AD method extracts the hyper-paramters of the SPIE-AD method so that atleast $r\%$ data from the training set falls within the robustness interval $[\rho_1, \rho_2]$, while minimizing $(\rho_2 - \rho_1)$. The algorithm currently is a brute force search through all possible hyper-parameter combination to find the best hyper-paramters that matched the above-mentioned conditions.

## A.5 DATA AND CODE AVAILABILITY

The data and code for model recovery using SINDY-MPC are available in `https://anonymous.4open.science/r/U2Recognition-5502/`

To use LTC-NN a manual transfer of model coefficient is required and the pipeline is not entirely automated. Hence, the models available in `https://anonymous.4open.science/r/LTC-NN-MR-4420/` has to be run first and the saved model coefficients needs to be transferred to the U2Recognition github and then run the files described in the U2Recognition github.

The AnomalySimpleton also known as SMDTrash is available in `https://anonymous.4open.science/r/AnomalyAbsurd-5CED/`

