# OpenReview forum: "Towards zero shot multivariate time series anomaly detection - A Realistic Evaluation"
_ICLR.cc/2025/Conference — Submitted to ICLR 2025_

### Official Review · Reviewer_awo8 · 2024-10-28

**Soundness:** 2
**Presentation:** 1
**Contribution:** 2
**Rating:** 3
**Confidence:** 4

**Summary:**

This paper introduces zero shot multivariate time series anomaly detection that can avoid point adjust step and data leakage in hyperparameter tuning. The proposed method is a sparse model recovery built upon ODE. A neural network is also used to handle the robustness. The proposed method could finally produce an interval based on the kth smallest value for labeling.

**Strengths:**

1. The proposed method can outperform existing methods by a large margin without point adjustment.
2. The hyperparameter of the proposed method can be tuned only with training data.
3. Build new benchmarks for better evaluating the U2 anomaly.

**Weaknesses:**

1. The writing needs a lot of improvement.  Below are some examples:
  + It is hard to follow the main content of the proposed method.
  + There are limited details about how the proposed benchmark can reflect the claimed properties.
  + Why can the claimed difficulty not be handled by some heuristics or advanced tech? For example, A1 could also be addressed by extreme value theory for dynamic threshold [1]. A2 could be mitigated by some better measurements to mitigate the effect of noisy labeling [2].

2. The experiments are not rigious, which decreases the confidence of the experiments.
  + The compared methods are too weak. In the real-world dataset evaluation, the compared method achieves a 0 F1 score. This is hard to trust. Why not provide heuristics like quantile for thresholding or extreme value theory for dynamic threshold?
  + Recent progress [2] in evaluating AD has proposed several threshold-free measurements. It seems the bad performance of the compared method largely results from only using F1 score, which requires a threshold.
  + There is zero empirical analysis for the proposed benchmark.
  + In Table 3, AT+ is the best but is not marked as bold in F8Slow.

[1] Anomaly detection in streams with extreme value theory. KDD 2017.
[2] Volume under the surface: a new accuracy evaluation measure for time-series anomaly detection. VLDB 2022.

**Questions:**

Please address the concerns in the weakness.

**Details Of Ethics Concerns:**

This paper proposes new data. There is a copyright concern.

---

> ### Author Response · Authors · 2024-11-16
> **Thank you for the review. Need a few clarifications to effectively respond**
>
> Dear reviewer,
>
> Can you kindly clarify what you mean by empirical analysis? How I interpreted empirical analysis is usage of real world data.
>
> The SMD, SMAP, and MSL datasets are real world datasets. Is that not empirical analysis?
>
>
> For ethics review part, all new data shared is simulation data and does not have any private identifiable information. In such a scenario, do you still have concerns with copyright?
>
> Please clarify so we can effectively address your comments.
>
> Thank you again for your insightful comments.

---

> ### Comment · Reviewer_awo8 · 2024-11-18
>
> 1. When a new benchmark is proposed, no matter whether synthetic data or real-world data, you want to have some distinct properties that existing benchmarks don't have. These properties have to be reflected by empirical analysis, especially for real-world datasets. This analysis should be convincing evidence, not just a simple quality analysis. We may expect statistical results for all the datasets about these claimed properties to be guaranteed, not just some samples. Also, it is better to see proof or discussion about these properties widely existing in the real world. Otherwise, It may be some ad hoc situations that may not be that important. The reason I didn't want to provide such a concrete description is that this is just one perspective. It is possible there could be another perspective to provide convincing evidence. I don't want a concrete description that involves some limitations to allow the authors to pursue potentially better directions.
>
> 2. Synthetic data doesn't mean you are allowed to publish these data and experimental results from these data. When you use real-world systems to generate something, the copyright is still a concern.

---

> ### Author Response · Authors · 2024-11-28
> **Revised version uploaded**
>
> 1.	Why cant A1 be handled by extreme value theory?
>
> The majority of benchmark techniques used in this paper use extreme value theory or peaks over threshold method discussed in Siffer et al 2017 paper. We have clarified this point in Figure 1. The threshold learning method is based on EVT and the POT algorithm is used in nearly all autoencoder based methods.
>
> EVT cannot handle A1 as evidenced in Table 3 and 4 where all baselines used EVT on anomaly score and performed poorly if point adjustment is removed.
>
> EVT can work if the anomaly data has a significant distribution shift from the normal data. In condition A3 we show that U2 data and normal data may not have a significant distribution shift invalidating the direct usage of EVT on data.
> The primary reason baselines use EVT is with the hope that the latent space representation of their model can potentially learn a transform that shows distribution shift between normal and anomalous data. As evidenced in this paper, baseline techniques apparently fail to learn such a latent space representation and hence have to take the help of point adjustment to artificially inflate precision.
>
> 2.	A2 could be mitigate by better measurements
>
> There is no relation of point adjustment with better measurements. Point adjustment is used in baseline techniques regardless of measurement quality.
>
> An argument could be made that quality measurements will lead to better anomaly definitions so that point adjustment will not be needed at all. But that is not a practical scenario, because we have to work with the sensing technology and data available to us. Better sensing will require more resources and better technology development. Until then anomaly detection techniques have to work with whatever measurements is available. Using point adjustment is unrealistic and cannot be used in a deployed system.
>
> 3.	The compared methods are too weak. In the real-world dataset evaluation, the compared method achieves a 0 F1 score. This is hard to trust.
>
> Anomaly transformer uses test data and labels as validation set. Please refer to the lines 196 to 200 in the dataloader.py code in the Anomaly Transformer github code. When we changed this code to use 20% of training data as validation set, the Anomaly Transformer F1 score was 0. Anomaly transformer is a transformer based method that uses a complex deep learning architecture and is by no means weak since it has been shown to outperform other baselines.
>
> To address your comments we have also included more sophisticated techniques such as TFAD Zhang et al 2022, FITS Xu et al 2024 and language model based OFA Zhou et al 2023 methods in Table 3 and 4. SPIE-AD is shown to outperform these SOTA techniques when point adjustment is not used.
>
> 4.	Recent progress [2] in evaluating AD has proposed several threshold-free measurements.
>
> The volume under surface metric is a threshold free measure. However, to evaluate VUS, we need the AUC-PR metric which is dependent on the threshold. SPIE-AD is a different class of anomaly detection technique that does not use an anomaly threshold. Hence, we cannot compute AUC-PR for different thresholds. As a result, we cant compute the VUS metric for SPIE-AD. For fair comparison we have thus used the best threshold settings for each baseline techniques that maximizes the point adjusted F1 score. This best threshold search method is implemented in every baseline and we did not tamper with it.
>
> 5.	There is zero empirical analysis for the proposed benchmark.
>
>  The main characteristics of our benchmark is that anomaly data and normal data does not have distribution shift. This is in line with real world cases such as Maneuvering Characteristics Augmentation System (MCAS) that controls the pitch of an aircraft. According to Herkert et al 2020, the MCAS was designed to mask any changes in flight characteristics between the older and newer versions of the Boeing aircraft. As a consequence if MCAS was designed correctly then it will nullify any distribution shift between MCAS operated flight and without MCAS flight. Now if an U2 scenario of wrongful MCAS trigger occurs, as in the fateful Lion Air or Ethiopean airlines, there will be no distribution change in the flight characteristics between U2 and normal scenario. Our aim is to replicate such real world U2 scenarios in simulation data. In Figure 1 we have shown that the synthetic data for the real world U2 scenarios and the normal data have same distribution. This is not only an experimental result but is proven using the Kolmogorov Smirnov test. Hence, our benchmarks have empirical basis. We have added this discussion in Introduction when we discuss A3.
>
> 6.	In Table 3, AT+ is the best but is not marked as bold in F8Slow.
>
> We have updated this in the revised paper. We did not want to highlight any performance that uses point adjustments. Hence we have removed bold markings on any method with a + in the updated table.

---

> ### Comment · Reviewer_awo8 · 2024-11-28
>
> Thanks for the responses. I believe these clarifications could largely improve the writing.
>
> Also, the responses cannot address the two critical concerns.
>
> 1. For compared methods, except for the latest methods, I also want to see you try some obvious tweaking to improve the existing methods and handle the issues. For example, why not have a thresholding that only considers time steps with top 5% anomaly scores and maybe have something similar to EVT or check the entropy within the 5% time steps. It seems there is still a gap in the density plot and it is likely to have something to directly utilize these observations.
>
> 2. Could your method output an anomaly score? If yes, the AUC-PR is still applicable by just considering the anomaly score. Even though you don't have a threshold, but you have some hyperparameter to control the top k smallest values, which is another type of threshold. Otherwise, I would like to see the detailed answer showing why AUC-PR cannot be computed.

---

> > ### Author Response · Authors · 2024-11-29
> >
> > Thank you for the interesting suggestions.
> >
> > 1. So if I understand correctly, what you are suggesting is that take the anomaly scores assigned by each method and instead of using threshold just take points that have top 5% of anomaly scores and mark them as anomalous. I can do that and show the results asap.
> >
> > 2. SPIE-AD does not give an anomaly score as output. It checks if a window consisting of the point of interest gives an underlying model that is statistically different from the other models. Now as shown in the new figure that we have added Figure 5 the accuracy varies with window size. We can definitely compute AUC-PR with respect to window size but not with respect to threshold. Hence the AUC-PR we can compute will be independent of threshold but dependent on window size. I will try to address these issues asap.

---

> ### Comment · Reviewer_awo8 · 2024-11-29
>
> 1. I just give an example. The 5% is not the exact thresholding. The number may depend on the percentage of the anomaly in each dataset. However, you didn't provide such important information. And, the 5% is not for labeling. It is more like the candidate instances. You may want additional selection with statistics in these candidate instances and label some of them as final anomaly. Also, what I mean is that you can tweak the existing methods and I only provide a possible direction, which may or may not be a strong one. But at least, it shows you can do something like this to make the compared methods more strong. You need to elaborate it, instead of depending on the reviewers.
>
> 2. Is the R_i not something relevant anomaly score? You have a threshold based on the variance d. Your proposed method is like, when the R_i is within the  radius d of c, all the relevant instances are normal. Otherwise, they are abnormal. So, doesn't mean R_i is the anomaly score for all the relevant instances? And yes, it is different from many existing anomaly score as it is not a monotonic value, but you can tweak it a little bit.

---

> ### Author Response · Authors · 2024-11-29
>
> Sorry for misunderstanding your first comment
>
> 1. I thought that you were asking us to fix the threshold at 5%. In my thinking that made sense because that will make the methods independent of the validation set and will violate A1. But your second comment clarified that you are asking a much more basic requirement that  attempts to vary different hyper-parameters such as quartiles of anomaly score and post processing to find the best performance.
>
> Actually this is already done in each of the methods. For example in anomaly transformer the explorations that you are referring to is performed in lines 253 to 325 in solver.py. All other autoencoder based techniques do that as well.
>
> In our results, we also did the same for all baselines. The results in Table 3 and 4 already have such explorations and we report with the threshold that results in the best performance.
>
> 2. The robustness interval is not a setting of our method. It is obtained from Theorem 1 and is determined from the training data. The setting that we can control is $\alpha$ which is the confidence measure. In all our experiments it is set to 0.05 which is standard in most statistical methods. As such $\alpha$ is not a threshold but it has a big role in determining the robustness range.
>
> To obtain AUC ROC and AUC PR, VUS ROC and VUS PR I varied this $\alpha$ from 0.05 to 0.2 in steps of 0.025. In this regard I used the code available in https://github.com/TheDatumOrg/VUS to compute all the abovementioned metrics of some of the baselines in Table 4. In particular I took the top 3 baselines and SPIE-ADS from Table 4 to get the table below.
>
> | **Method**          | **SMD AUC ROC** | **SMD AUC PR** | **SMD VUS ROC** | **SMD VUS PR** | **SMAP AUC ROC** | **SMAP AUC PR** | **SMAP VUS ROC** | **SMAP VUS PR** | **MSL AUC ROC** | **MSL AUC PR** | **MSL VUS ROC** | **MSL VUS PR** |
> |----------------------|------------|--------------|-------------|--------------|--------------|--------------|--------------|--------------|-------------|-------------|-------------|-------------|
> | **AT**              | 0.48       | 0.46         | 0.59           | 0.56         | 0.37         | 0.35         | 0.44            | 0.41          | 0.29        | 0.25        | 0.33           |0.29         |
> | **FITS**            | 0.57       | 0.55         | 0.62        | 0.59          | 0.41        | 0.37         | 0.47          | 0.43          | 0.42       | 0.38        | 0.47         | 0.44         |
> | **TFAD**            | 0.6       | 0.55         | 0.64         | 0.58          | 0.43         | 0.41         | 0.47          | 0.44          | 0.46        | 0.44        | 0.53         | 0.5         |
> | **SPIE-ADS**        | 0.61         | 0.58           | 0.65          | 0.61          | 0.52           | 0.51           | 0.57           | 0.54          | 0.62          | 0.58          | 0.64          | 0.59         |

---

> > ### Author Response · Authors · 2024-12-03
> >
> > Dear reviewer,
> >
> > Thank you for interacting during the process. We have addressed your latest comments as well with more experiments. Hope you have gotten a chance to check our response. Please let us know if you seek any more clarifications.

---

### Official Review · Reviewer_sZLJ · 2024-10-30

**Soundness:** 3
**Presentation:** 2
**Contribution:** 3
**Rating:** 5
**Confidence:** 4

**Summary:**

The paper proposes a sparse model identification enhanced anomaly detection (SPIE-AD) model which is a recovery and conformance-based zero shot MTAD approach. It disables point adjustment and thresholding methods. The backbone of SPIE-AD are the two fundamental theoretical contributions of this paper: a) robust sparse non-linear dynamical model recovery from real-world multi-variate data using neural architectures with automated differentiation and b) statistical conformance-based model robustness interval extraction (CRIE) algorithm that can identify statistically relevant difference in recovered models.

**Strengths:**

The proposed model using dynamic modeling is novel. And paper is mathematically sound and well-written. Performance on synthetic data is good.

**Weaknesses:**

U2 scenario is nonrealistic as real world data is mixed with unknown noise. It is hard to get training data cleaned up in real-world data. It will be better to discuss how author would handle noisy or imperfect training data in real-world applications.

What is a contribution against the Autoencoder-based method? Can you please compare SPIE-AD approach to autoencoder-based methods like USAD, highlighting key differences and potential advantages in terms of performance, computational efficiency, or interpretability.

Also, validation data should not be used as training data. Is there any real-world application in your mind behind this?

What happens if you use separate validation data? It would be better to conduct an additional experiment using a separate validation set that is distinct from both the training and test sets and kindly report how this affects the results.

The author models robustness as a reconstruction error. Why the dynamic model is called sparse and robust in not clear. Please provide a more detailed explanation of the definition of "sparse" and "robust" in the context of their dynamic model. Can you please explain how sparsity is enforced in the model structure and how robustness is quantified beyond reconstruction error?  Some experimental evidence on the usefulness of sparsity and robustness for detecting anomalies would be nice to have.


Table 4 shows that the proposed model is performing worse than many benchmarks. Please provide a more in-depth analysis of these results, discussing potential reasons for underperformance.

Some analytical results about where the proposed method fails would be good.  Please provide a detailed error analysis, identifying specific types of anomalies, datasets, or scenarios where your method underperforms. Please also discuss potential reasons for these failures and propose potential improvements or future work to address these limitations.

**Questions:**

As above.

---

> ### Author Response · Authors · 2024-11-25
> **Response to your comments**
>
> Thank you, very insightful points were raised.
> 1. U2 scenario is nonrealistic
>
> In autonomous systems, U2 are scenarios that never occur during design but occur during deployment. This is a very realistic scenario and occurs in many autonomous systems.
> Realistic U2 scenarios are phantom braking in Tesla, where the car stops in a free way without traffic;  Zero day network attacks, where a vulnerability did not exist in design but occurred in test and hence not available in training data. Autonomous cars, Waymo, going wrong way, which was never observed during design. Unknown noise are stochastic non-causal variance and are different from U2 scenario, which are causal events. We do not clean up training data by removing anomalous data. U2 data are by definition not available in the training data.
>
> 2. Handling noisy/imperfect training data
>
> We have three real world datasets SMAP, MSL and SMD. We deal with imperfect training data and are still able to beat SOTA anomaly detection methods without the need for using point adjustment to artificially inflate precision.
>
> 3. Contributions beyond autoencoders
>
> A. Autoencoder based methods are purely data driven and rely on learning an accurate latent space representation of the data.
> SPIE-AD tries to extract the underlying governing equations which is generating the data.
>
> B. Autoencoder based methods generate an anomaly score from the latent space and use extreme value theory to determine anomalies. They only learn the latent space representation once and apply it to test data.
> SPIE-AD recognizes anomalies by monitoring significant difference in underlying model of the data. It continually extracts the model from the data and compares with existing model
>
> C. Nearly all autoencoder based models use point adjustment (PA) to inflate precision. Without PA autoencoder based methods have very poor precision as shown in Table 3.
> SPIE-AD does not use PA. As a result SPIE-AD shows poorer precision than autoencoder based methods with PA. However, without PA, SPIE-AD outperforms autoencoder based methods with a significant margin. Thus, SPIE-AD is applicable in practical deployments where the ground truth is not available to implement PA.
>
> D. Autoencoder based methods use high dimensional latent space representation
> SPIE-AD focuses on recovering sparse models with fewest parameters and is computationally and storage wise more efficient.
>
> 4. Validation vs. training data
>
> We use a part of training data (20%) as validation data and keep it separate from training (rest of 80%) and test. We only use validation data for early stopping. On the other hand SOTA autoencoder based methods use test data as validation data. This causes significant data leakage and is not a good practice in machine learning.
>
> 5. Sparse and robust dynamic model
>
> Sparsity is used to ensure that we learn the best possible model with the least number of parameters. This is guided by Occam’s Razor theorem that tells us to chose the model with the least number of parameters for better generalization if two models have same performance. The sparsity is induced by setting dense layer outputs that are below a certain low threshold (<1% of the mean value of all dense layer outputs) to 0. Our main aim for robustness is to reconstruct the original data under high noise and low sampling rate. In order to ensure robustness, we replaced the SINDY-MPC core of SPIE-AD to a liquid time constant neural network (Figure 4). LTC-NN networks are more suitable for robust model learning (Hasani et al 2024).
>
> 6. Performance in Table 4
>
> Table 4 shows that in order to outperform SPIE-AD SOTA techniques require PA. If we remove PA (not A2 column), SOTA performance drops drastically and SPIE-AD outperforms all SOTA. If validation set is kept separate from test data then SOTA techniques have more performance degradation.
> Further, as pointed out in public comments, the ground truth labelling of SMD, SMAP, and MSL may be wrong. This might be another reason that SPIE-AD does not perform as well. We do not want to use PA because as also pointed out by Kim et al in AAAI 2022, PA unfairly inflates precision based on the length of anomaly. PA is unrealistic and cannot be used in real world. We should only compare (not A1 or not A2 column) in Table 4 which clearly shows that SPIE-AD outperforms SOTA methods by a significant margin.
>
> 7. Analytical results on method limitations
>
> The most challenging anomalies for our method are small anomalies that have very less number of samples. If we have a window size of W and an overlap of W-1 then the performance of our technique improves significantly. However, an overlap of W-1 is computationally prohibitive. The reported results are with a window overlap of 50% which is shown to outperform SOTA techniques when we do not use PA. If the normal data has significant noise then also it becomes difficult to identify anomalies. We improve robustness by changing the SINDY core to LTC-NN network.

---

> > ### Author Response · Authors · 2024-11-28
> > **Revised paper uploaded**
> >
> > Thank you for your patience. We have made some changes to the paper following your suggestion.
> >
> > 1.	U2 scenario is nonrealistic
> >
> > We have provided an additional example of realistic U2 case. We have discussed the case of erroneous trigger of maneuvering characteristics augmentation system (MCAS) that resulted in the crash of Lion Air and Ethiopean airlines. In this case, the erroneous triggering of MCAS was not seen during design time and hence was not available in training data. This example in discussed in Introduction section.
> >
> > 2.	Handling noisy/imperfect training data
> >
> > We have three real world datasets SMAP, MSL and SMD. We have added 250 new real world datasets available in UCR database for anomaly detection in our evaluation as well. This is highlighted in Section 5.3 and in Table 2. We deal with imperfect training data and are still able to beat SOTA anomaly detection methods without the need for using point adjustment (PA) to artificially inflate precision.
> >
> > 3.	Contributions beyond autoencoders
> >
> > A. Autoencoder based methods are purely data driven and rely on learning an accurate latent space representation of the data. SPIE-AD tries to extract the underlying governing equations which is generating the data.
> >
> > B. Autoencoder based methods generate an anomaly score from the latent space and use extreme value theory to determine anomalies. They only learn the latent space representation once and apply it to test data. SPIE-AD recognizes anomalies by monitoring significant difference in underlying model of the data. It continually extracts the model from the data and compares with existing model
> >
> > C. Nearly all autoencoder based models use PA to inflate precision. Without PA autoencoder based methods have very poor precision as shown in Table 3. SPIE-AD does not use PA. As a result SPIE-AD shows poorer precision than autoencoder based methods with PA. However, without PA, SPIE-AD outperforms autoencoder based methods with a significant margin. Thus, SPIE-AD is applicable in practical deployments where the ground truth is not available to implement PA.
> >
> > D. Autoencoder based methods use high dimensional latent space representation SPIE-AD focuses on recovering sparse models with fewest parameters and is computationally and storage wise more efficient.
> >
> > 4.	Validation vs. training data
> >
> > We use a part of training data (20%) as validation data and keep it separate from training (rest of 80%) and test. We only use validation data for early stopping. On the other hand SOTA autoencoder based methods use test data as validation data. This causes significant data leakage and is not a good practice in machine learning.
> >
> > 5.	Sparse and robust dynamic model
> >
> > Sparsity is used to ensure that we learn the best possible model with the least number of parameters. This is guided by Occam’s Razor theorem that tells us to chose the model with the least number of parameters for better generalization if two models have same performance. The sparsity is induced by setting dense layer outputs that are below a certain low threshold (<1% of the mean value of all dense layer outputs) to 0. Our main aim for robustness is to reconstruct the original data under high noise and low sampling rate. In order to ensure robustness, we replaced the SINDY-MPC core of SPIE-AD to a liquid time constant neural network (Figure 4). LTC-NN networks are more suitable for robust model learning (Hasani et al 2024).
> >
> > 6.	Performance in Table 4
> >
> > Table 4 shows that in order to outperform SPIE-AD SOTA techniques require PA. If we remove PA (not A2 column), SOTA performance drops drastically and SPIE-AD outperforms all SOTA. If validation set is kept separate from test data then SOTA techniques have more performance degradation. Further, as pointed out in public comments, the ground truth labelling of SMD, SMAP, and MSL may be wrong. This might be another reason that SPIE-AD does not perform as well. We do not want to use PA because as also pointed out by Kim et al in AAAI 2022, PA unfairly inflates precision based on the length of anomaly. PA is unrealistic and cannot be used in real world. We should only compare (not A1 or not A2 column) in Table 4 which clearly shows that SPIE-AD outperforms SOTA methods by a significant margin.
> >
> > 7.	Analytical results on method limitations
> >
> > We have added new experiments on the UCR real world anomaly detection database to evaluate sensitivity of SPIE-AD on window size. We have added a new limitations section in Section 6.

---

> > > ### Author Response · Authors · 2024-12-03
> > >
> > > Dear Reviewer,
> > >
> > > We could not really interact during the process. Hope you have gotten a chance to check the revised version. We have attempted to incorporate your suggestions. Please let us know if you need any more clarifications.

---

### Official Review · Reviewer_T8KH · 2024-11-03

**Soundness:** 3
**Presentation:** 2
**Contribution:** 2
**Rating:** 5
**Confidence:** 3

**Summary:**

This paper proposes a multi-variant time series anomaly detection (MTAD) method that differs from existing approaches by learning the relationships between points on a curve. Instead, it judges whether a certain time series is anomalous by comparing the changes in model parameters \( w_i \) fitted from different data. This method no longer relies on labels from the validation set to tune the model and can achieve good results without using point adjustment.

**Strengths:**

1. The paper points out that both point adjustment and reliance on validation labels are significant issues.
2. The application of the SMR method to anomaly detection in the paper is innovative.
3. The paper contributes a synthetic evaluation dataset.

**Weaknesses:**

1. More related methods should be compared and discussed.
2. The writing part of the paper requires some improvement; the introduction is too long.
3. The references are not placed in parentheses, which affects readability.
4. The placement of related work in the middle of the paper, together with the baseline in the evaluation, is not intuitive. Related work should be an independent section as it discusses relevant papers in this research field.

**Questions:**

1. The paper uses relatively simple baselines. Some methods with better performance, such as TFAD[1], OFA[2], and FITS[3], are not compared. It is uncertain whether the method in this paper still has certain advantages under zero-shot evaluation methods and the u2 dataset.
2. The paper only mentions the SMD dataset, but there is a benchmark[4] that even focuses on zero-shot and point adjustment alternatives. How does the paper's method perform on this benchmark?

[1] TFAD: A decomposition time series anomaly detection architecture with time-frequency analysis

[2] Zhou T, Niu P, Sun L, et al. One fits all: Power general time series analysis by pretrained lm[J]. Advances in neural information processing systems, 2023, 36: 43322-43355.

[3] FITS: Modeling time series with 10k parameters, in The Twelfth International Conference on Learning Representations, 2024.

[4] Si, Haotian, et al. "Timeseriesbench: An industrial-grade benchmark for time series anomaly detection models." arxiv.org/pdf/2402.10802 (ISSRE 2024).

---

> ### Author Response · Authors · 2024-11-22
> **Revision plan**
>
> Dear Reviewer,
>
> Thank you very much for your insightful comments. We are planning on addressing your comments in the revised version of the paper as follows:
>
> a) We are including the OFA, TFAD and FITS as comparators in our analysis.
>
> All three approaches use point adjustment approach. Please check lines 313 to 331 in solver_recon.py for FITS, line 186 in exp_anomaly_detection.py in OFA, and in line 438 in tfad.py in the TFAD approach.
>
> When we comment out these parts of the code to avoid using point adjustment, the Precision for OFA falls from 89% to 20% on an average and recall falls from 82% to 4%. Similar drops in precision and recall for FITS and TFAD are also observed.
>
> Our core hypothesis also supported by Kim et al AAAI 2022 is that usage of point adjustment unfairly adjusts true positives and inflates Precision and is an unrealistic evaluation. That is why we want to evaluate all techniques without point adjustment. We will include these three benchmarks in our results table.
>
> b) The main problem that we solve in the paper is multi-variate time series anomaly detection. Timeseriesbench is univariate time series anomaly detection. Following your suggestion, we have applied our technique on univariate time series anomaly detection as well. In the interest of space, we plan to have comparison results with Timeseriesbench in the supplementary document.
>
> c) We didnt just discuss SMD dataset. In our paper, we discuss three real world benchmark datasets SMD, SMAP and MSL in addition to 4 U2 datasets. Please refer to Table 2 and Table 4 in the paper.
>
> d) We will address all writing issues highlighted in your review.
>
> Thank you for your thorough review,
>
> We will soon upload the revise paper. In the meantime if you have further comments, please let us know.

---

> > ### Author Response · Authors · 2024-11-28
> > **Revised paper uploaded**
> >
> > Thank you for your patience. Here is a summary of changes we made to our paper.
> >
> > a)	Use TFAD, FITS and OFA as baselines
> >
> > In Table 3 and 4 we have now added TFAD, FITS and OFA as well as OmniAnomaly. TFAD, FITS, and OFA performance drastically drops when PA is removed and SPIE-AD outperforms them.
> >
> > b)	Use Timeseriesbench as a benchmark
> >
> > We have included the UCR database, which is the largest dataset used in Timeseriesbench. Timeseriesbench also uses Yahoo dataset. However, as pointed out in public comment and also in “Current Time Series Anomaly Detection Benchmarks are Flawed and are Creating the Illusion of Progress” by Wu et al. IEEE TKDE 2023 that Yahoo dataset has mislabeling errors. Hence, we used the UCR database that consists of 250 real world anomaly detection datasets. The results are summarized in Section 5.3 and also in Figure 5.
> >
> > c)	The paper only mentions the SMD dataset
> >
> > We didn’t just discuss SMD dataset. In our paper, we discuss three real world benchmark datasets SMD, SMAP and MSL in addition to four U2 datasets. Please refer to Table 2 and Table 4 in the paper.
> >
> > d)	Writing issues
> >
> > We now have a separate related work section and a baseline technique section. We have also added a discussion on TFAD, FITS, OFA in the related work section.

---

> > > ### Author Response · Authors · 2024-12-03
> > >
> > > Dear Reviewer,
> > >
> > > We were able to incorporate the changes that you had suggested. Hope you have gotten a chance to check the revised paper. Please let us know if you seek any further clarification.

---

### Official Review · Reviewer_V6nE · 2024-11-03

**Soundness:** 3
**Presentation:** 3
**Contribution:** 3
**Rating:** 8
**Confidence:** 4

**Summary:**

The paper introduces SPIE-AD, a zero-shot multivariate time series anomaly detection approach that aims at overcoming the prevalent information leakage from threshold selection prevalent in the literature. The key innovation is combining sparse model identification with conformance testing to detect anomalies without requiring anomaly examples during training. The authors also contribute new synthetic benchmarks and demonstrate issues with current evaluation practices.

**Strengths:**

The paper tackles an important problem, pervasive in the anomaly detection literature that puts into question the validity of a lot of published results in real-work scenarios. The paper introduces a novel technical approach combining model identification with conformance testing and performed a comprehensive empirical evaluation including with the release of new benchmarks. The theoretical foundations of the paper appear sound.

**Weaknesses:**

The paper contains limited discussion of computational complexity of the proposed methods. This is an important weakness given that the proposed method is not straightforward. Similarly, the authors could better explain parameter sensitivity and failure cases.
The issue with information leakage from calibration of threshold on the training set has been identified in previous works, the related literature section is missing a discussion of this.

The empirical analysis is limited. It relies mostly on the synthetic datasets proposed by the authors. The quality and informativeness of the real-world datasets they use have been questioned, rightfully so in this reviewer's opinion. UCR offers a large collection on time-series anomaly datasets that are well documented. The authors should use them. Similarly, the authors should include simple, algorithmic baselines to their comparison and include computational cost as a relevant dimension.

**Questions:**

What is the computational overhead of SPIE-AD compared to existing methods?
How sensitive is the method to the choice of window size?
Are there specific types of anomalies where the method might fail?

---

> ### Author Response · Authors · 2024-11-26
> **Response to your comments**
>
> Thank you for your comments. Here is our response and revision plan.
>
> 1. Computational complexity
> SPIE-AD has two model recovery cores: a) SINDY-MPC and b) LTC-NN.
>
> SINDY-MPC uses the sequential threshold ridge regression strategy. The computational complexity of Ridge regression in the worst case is O($Nn^2$), where $N$ is the number of samples and $n$ is the dimension of the multivariate signal if the number of regularization parameters is less than $N$, which is the case in the example of anomaly detection. The sequential threshold runs Ridge regression multiple times until a desired reconstruction accuracy is obtained. If we fix a maximum $Q$ number of times that the sequential threshold can run then the overall computational complexity of SINDY-MPC is O($QNn^2$).
>
> For the LTC-NN architecture, the computation complexity of forward pass is $O(V+V\times (|\Theta|+q)) + O(|X|N)$, where $N$ is the number of samples in the data, $V$,$q$,$\Theta$,$X$ are described in Figure 4. The complexity of backward pass is $O(V\times P_{LTC}\times N + V \times (|\Theta|+q) \times P_{dense} \times N)$, where $P_{LTC}$ is the number of parameters in the LTC cell, and $P_{dense}$ is the number of parameters in each neuron of the dense layer. SINDY-MPC ran on a single CPU thread and was 11.3 ($\pm$ 2.1) times faster than the neural architecture on GPU.
>
> 2. sensitivity to window size
>
> The SINDY-MPC strategy in SPIE-AD is very sensitive to choice of window size. This is also a drawback of SINDY-MPC highlighted in Chen at al https://www.nature.com/articles/s41467-021-26434-1. However, it can be addressed very well by the LTC-NN architecture which has low sensitivity to window size. We are including one experiment to show this in the revised paper.
>
> 3. anomalies where SPIE-AD might fail
>
> Our technique faces challenges in determining point anomalies that are very short lasting few samples. From what we have observed in the SMD SMAP and MSL datasets, anomalies that last < 5 samples are missed consistently.
>
> 4. Use of UCR dataset
> Following your comments, we have implemented on UCR dataset (250 datasets). We use the accuracy metric as described in https://arxiv.org/pdf/2311.12550v5.
>
> | **Method**           | **Mechanism**                           | **Published**      |     **Top-1 Acc.** |
> |-----------------------|-----------------------------------------|------------------------|----------------|
> | non-DL               | one-class classification               | OC-SVM [33]                | 0.088          |
> | non-DL               | isolation forest                       | IF [3]                 | 0.376          |
> | non-DL               | isolation forest                       | RCF [4]                  | 0.387          |
> | non-DL               | matrix profile                         | Matrix Profile SCRIMP [34] | 0.416          |
> | non-DL               | density estimation                     | MDI [13]                 | 0.47           |
> | non-DL               | matrix profile                         | Matrix Profile STUMPY [35]  | 0.512          |
> | non-DL               | discord discovery                      | MERLIN [5]                 | 0.424          |
> | non-DL               | discord discovery                      | MERLIN++ [6]             | 0.424          |
> | DL                   | reconstruction                         | AE                    | 0.236          |
> | DL                   | reconstruction                         | Convolutional AE         | 0.352          |
> | DL                   | reconstruction                         | LSTM-ED [36]            | 0.51           |
> | DL                   | variational reconstruction             | LSTM-VAE [8]        | 0.198          |
> | DL               | forecasting                            | Telemanom [11]             | 0.468          |
> | DL                   | one-class classification               | Deep SVDD [2]       | 0.076          |
> | DL             | density estimation                     | DAGMM [14]             | 0.061          |
> | DL             | spectral saliency map                  | SR-CNN [56]          | 0.30           |
> | DL                   | reconstruction, adversarial training   | USAD [9]         | 0.276          |
> | DL                   | contrastive learning                   | CPC-AD [40]           | 0.064          |
> | DL          | contrastive learning, one-class classification | TS-TCC-AD [57, 58] | 0.006          |
> | DL              | reconstruction                         | TranAD [10]               | 0.19           |
> | DL        | density estimation                     | GANF [15]                | 0.24           |
> | DL       | non-contrastive learning               | COCA [20]            | 0.236          |
> | DL      | density estimation                     | TimeVQVAE-AD [1]       | 0.708          |
> |**no DL** | **SPIE-AD** | **This paper**  |**0.756**  |
>
> All references are available in [1]: https://arxiv.org/pdf/2311.12550v5

---

> ### Author Response · Authors · 2024-11-28
> **Revision uploaded**
>
> Thank you for your patience. Here is a summary of changes we made to our paper.
>
> 1.	Computational cost
>
> We have added an analytical exploration of computational cost of SPIE-ADS and SPIE-ADL in Section 2.5 of the paper. In addition, we have also included execution time of SPIE-AD and its comparators in Table 4 of the revised paper.
>
> 2.	Sensitivity to window size
>
> We have added a new experiment on the UCR database to evaluate sensitivity of our method to window size. We have added Figure 5 that shows the variation of average accuracy of the SPIE-AD technique on window size for 250 real world datasets available in the UCR database.
>
> 3.	Information leakage from calibration of threshold
>
> This comes from the Peaks over threshold extreme value theory mechanism discussed in Siffer et al. 2017 KDD paper. The original method advocated for using the EVT on validation data, that is obtained from the training data. However, as also supported by Wu and Keog et al 2023, the baseline techniques use EVT on test data and then evaluate performance on the same test data leading to data leakage. We have updated the discussion of A1, A2 and A3 by citing relevant research in Introduction section. We have also updated Figure 1 to reflect this detail.
>
> 4.	Use of UCR dataset
>
> We have added performance evaluation of our technique on UCR database. This is in Table 2, and Section 5.3. As compared to the leaderboard published for the UCR database SPIE-AD beats the SOTA techniques by 4.8%. The SOTA in the leaderboard published for UCR database includes algorithmic baselines such as Matrix Profile and USAD.
>
> Furthermore, in the multi-variate real world datasets of SMD, SMAP, and MSL, we have added four additional baselines: a) algorithmic baseline of TFAD and FITS that depends on time frequency analysis of the multivariate data, and b) language model based technique of one size fits all (OFA). This is highlighted in the results table Table 3 and 4.

---

> > ### Author Response · Authors · 2024-12-03
> >
> > Dear Reviewer,
> >
> > We didn’t really get a chance to interact. Hope you have gotten a chance to read the revised version. Please let us know if you need any further clarifications.

---

### Official Review · Reviewer_fJ9Q · 2024-11-04

**Soundness:** 3
**Presentation:** 2
**Contribution:** 2
**Rating:** 5
**Confidence:** 4

**Summary:**

This paper proposes a zero-shot time series anomaly detection method called SPIE-AD, which leverages sparse model recovery (SMR) technology to detect anomalies by monitoring changes in model parameters. SPIE-AD makes two main modifications: it does not rely on manually labeled labels for hyperparameter tuning in the validation set, and it abandons the controversial point adjustment evaluation method that has been heavily relied upon. The evaluation results are explicit, showing that SPIE-AD significantly outperforms existing methods on the U2 benchmark.

**Strengths:**

1. The method proposed in the paper for anomaly detection through model recovery is innovative, as traditional methods that model normal patterns based on data and then compare them have high dependencies on the quantity and quality of data.
2. The challenge of consistent distribution between anomalous and normal data in the U2 benchmark presented in this paper, which is difficult to distinguish, has not been addressed in many works, making it somewhat innovative.

**Weaknesses:**

1. The meanings of H and P in Figure 3 are not indicated.
2. Lines 6 and 9 in Algorithm 2 are reversed.
3. On line 84, it is stated that existing anomaly detection methods are based on the assumption of anomaly-free data, which is invalid. The biggest difference between anomaly detection tasks and time series tasks is that anomaly detection tasks do not assume that training data is free of anomalies, which is why there are many reconstruction methods have been proposed.
4. The relationship between zero-shot and A1, A2, A3 needs to be clarified. Are they parallel, or do A1, A2, and A3 make the zero-shot objective face greater challenges?

**Questions:**

1. Some very classic MTS evaluation datasets, besides SMD, have not been evaluated, such as Yahoo[1].
2. Some very classic MTS baselines and methods have not been compared, such as OmniAnomaly[2].
3. To achieve good zero-shot performance, some time series foundation models proposed through time series analysis have been proposed, which can do both time series forecasting and anomaly detection. These methods should also be compared, such as OFA's sec 4.2[3], and some related work mentioned by OFA.
4. A very similar evaluation work, TimeseriesBench[4], has not been mentioned. TimeseriesBench discusses zero-shot and point adjustment comparisons. How would SPIE-AD perform on TimeseriesBench?

[1] Y. Research, “A benchmark dataset for time series anomaly detection.” https://yahooresearch.tumblr.com/post/114590420346/a-benchmark-dataset-for-time-series-anomaly, 2015.
[2] Su, Ya, et al. "Robust anomaly detection for multivariate time series through stochastic recurrent neural network." Proceedings of the 25th ACM SIGKDD international conference on knowledge discovery & data mining. 2019.
[3] Zhou, Tian, et al. "One fits all: Power general time series analysis by pretrained lm." Advances in neural information processing systems 36 (2023): 43322-43355.
[4] Si, Haotian, et al. "Timeseriesbench: An industrial-grade benchmark for time series anomaly detection models." arXiv preprint arXiv:2402.10802 (2024).

---

> ### Author Response · Authors · 2024-11-22
> **Revision Plan**
>
> Thank you very much for your comments on our paper. Here is the plan for revising the paper according to your comments:
>
>
> 1. The meanings of H and P in Figure 3 are not indicated.
>
> Answer: H is a boolean variable that indicates whether the two distributions in blue and red dashed lines are statistically equivalent (H = 0) or not (H = 1) according to the KS test. P is a real variable that gives the probability of making a Type 1 error in rejecting the null hypothesis that the two distributions are indeed equivalent (H = 0). A low p value means the null hypothesis is rejected i.e. (H = 1) with very low error.
>
> As seen in Figure 3, for the standard anomaly detection datasets, the distributions of anomalous data and normal data are statistically very different hence rejecting the null hypothesis. This is indicated by H = 1 and a very low p value. On the other hand for U2 examples, the U2 data and the normal data distributions are nearly equivalent (second row in Figure 3). This is indicated by H = 0 and a high p value close to 1. However, as seen in third row, if we consider the underlying model connecting the multiple variables in the dataset, we see significant difference in the distribution of the model coefficients of the underlying model (H = 1 and p nearly 0). This forms the basis of our approach, where we learn the underlying dynamics connecting the multiple variates of the data and then try to find difference in model parameters of normal and U2 data.
>
> We agree that the Figure 3 needs to be better explained with clear definition of H and P and are updating in the revised paper.
>
>
>
> 2. Lines 6 and 9 in Algorithm 2 are reversed.
>
>
> Answer: We agree we will update this in the revised version.
>
>
> 3. existing anomaly detection methods are based on the assumption of anomaly-free data,
>
>
> Answer: We made a inadvertent mistake. We meant to say that the training data does not have anomaly labels. This is highlighted in Figure 1, where we clearly say that for training there are no labels. We will update this in the revised paper.
>
>
> 4. A1 A2 and A3 and their relationship with zero shot
>
>
> Answer: Zero shot MTAD is required for U2 detection because U2 errors by definition do not occur in training set. A1 A2 and A3 make the zero shot MTAD more challenging. We will clearly specify this in the revised paper.
>
>
> 5. Some very classic MTS evaluation datasets, besides SMD, have not been evaluated, such as Yahoo[1].
>
>
> 6.  very similar evaluation work, TimeseriesBench[4], has not been mentioned. TimeseriesBench discusses zero-shot and point adjustment comparisons. How would SPIE-AD perform on TimeseriesBench?
>
>
> Answer: Our main aim is multi-variate time series anomaly detection. To the best of our knowledge and investigation, the Yahoo dataset  is univariate as also highlighted in the TFAD paper (Table 1) https://arxiv.org/pdf/2210.09693.
>
> Based on your suggestion, we have implemented SPIE-AD on univariate datasets as well including Yahoo, TimeseriesBench and some of the UCR datasets (https://paperswithcode.com/dataset/ucr-anomaly-archive) as well. However, both TImeseriesBench and UCR datasets are also univariate datasets.
>
> In the interest of space, we intend to put all comparison with univariate timeseries datasets into supplementary documents.
>
>
>
> We are currently revising the paper and will soon upload the revised version. If you have any further issues you would like us to address please let us know. Thank you again for your valuable comments.

---

> > ### Comment · Reviewer_fJ9Q · 2024-11-26
> >
> > Thank you for the author's reply. The zero-shot setup and the point-adjustment issues proposed in this paper are commendable, and there was a previous commitment to update the latest version. However, after carefully reviewing the latest PDF, I saw no updates to the experiments.  Additionally, I have a minor issue. There seems to be a formatting problem with the authors' response, and the display on the OpenReview webpage looks rather odd. Could you please fix the Markdown syntax?

---

> > > ### Author Response · Authors · 2024-11-26
> > > **Still updating**
> > >
> > > Dear Reviewer,
> > >
> > > We are still working on the revision. We plan to update a version by the 27 th since iclr extended their deadline. We will upload a new version addressing all reviewer comments.
> > >
> > > Thank you for your patience and interacting with us.

---

> > > > ### Author Response · Authors · 2024-11-26
> > > > **Snippet of results**
> > > >
> > > > Dear reviewer,
> > > >
> > > > While we are still updating our paper here is a sneak peak at our results. We have tested our technique on the UCR dataset. It consists of 250 datasets. Below is one table we are adding in the revised paper.
> > > >
> > > > | **Method**           | **Mechanism**                           | **Published**      |     **Top-1 Acc.** |
> > > > |-----------------------|-----------------------------------------|------------------------|----------------|
> > > > | non-DL               | one-class classification               | OC-SVM [33]                | 0.088          |
> > > > | non-DL               | isolation forest                       | IF [3]                 | 0.376          |
> > > > | non-DL               | isolation forest                       | RCF [4]                  | 0.387          |
> > > > | non-DL               | matrix profile                         | Matrix Profile SCRIMP [34] | 0.416          |
> > > > | non-DL               | density estimation                     | MDI [13]                 | 0.47           |
> > > > | non-DL               | matrix profile                         | Matrix Profile STUMPY [35]  | 0.512          |
> > > > | non-DL               | discord discovery                      | MERLIN [5]                 | 0.424          |
> > > > | non-DL               | discord discovery                      | MERLIN++ [6]             | 0.424          |
> > > > | DL                   | reconstruction                         | AE                    | 0.236          |
> > > > | DL                   | reconstruction                         | Convolutional AE         | 0.352          |
> > > > | DL                   | reconstruction                         | LSTM-ED [36]            | 0.51           |
> > > > | DL                   | variational reconstruction             | LSTM-VAE [8]        | 0.198          |
> > > > | DL               | forecasting                            | Telemanom [11]             | 0.468          |
> > > > | DL                   | one-class classification               | Deep SVDD [2]       | 0.076          |
> > > > | DL             | density estimation                     | DAGMM [14]             | 0.061          |
> > > > | DL             | spectral saliency map                  | SR-CNN [56]          | 0.30           |
> > > > | DL                   | reconstruction, adversarial training   | USAD [9]         | 0.276          |
> > > > | DL                   | contrastive learning                   | CPC-AD [40]           | 0.064          |
> > > > | DL          | contrastive learning, one-class classification | TS-TCC-AD [57, 58] | 0.006          |
> > > > | DL              | reconstruction                         | TranAD [10]               | 0.19           |
> > > > | DL        | density estimation                     | GANF [15]                | 0.24           |
> > > > | DL       | non-contrastive learning               | COCA [20]            | 0.236          |
> > > > | DL      | density estimation                     | TimeVQVAE-AD [1]       | 0.708          |
> > > > |**no DL** | **SPIE-AD** | **This paper**  |**0.756**  |
> > > >
> > > > All references are available in [1]: https://arxiv.org/pdf/2311.12550v5

---

> > > > > ### Author Response · Authors · 2024-11-28
> > > > > **Revision uploaded**
> > > > >
> > > > > Thank you very much for your comments on our paper. Here we summarize our changes to address your comments.
> > > > >
> > > > > 1.	The meanings of H and P in Figure 3 are not indicated.
> > > > >
> > > > > Answer: We have updated the description of Figure 3 and elaborated meaning of H and P in line 136 to 145 of revised paper.
> > > > >
> > > > > 2. Lines 6 and 9 in Algorithm 2 are reversed.
> > > > >
> > > > > Answer: We have updated this in Algorithm 2.
> > > > >
> > > > > 3. existing anomaly detection methods are based on the assumption of anomaly-free data,
> > > > >
> > > > > Answer: We updated this in Figure 1 and also corrected in line 50-51.
> > > > >
> > > > > 4. A1 A2 and A3 and their relationship with zero shot
> > > > >
> > > > > Answer: We have updated description of A1, A2 and A3 in Introduction to make the point that A1 and A2 are problems with existing MTAD, while A3 is a characteristic of real world U2 scenarios that makes the zero shot MTAD more challenging. We have used the maneuvering characteristics augmentation system (MCAS) error that crashed Ethiopean airlines and Lion air flights as an example of real life U2 error and justified A3. MCAS was designed to mask the distribution difference between normal flight and stalled flight. Hence erroneous triggering of MCAS would not have been identified by a distribution change in the flight parameters. This is highlighted in lines 110 to 114.
> > > > >
> > > > > 5. Some very classic MTS evaluation datasets, besides SMD, have not been evaluated, such as Yahoo [1].
> > > > > 6. very similar evaluation work, TimeseriesBench[4], has not been mentioned. TimeseriesBench discusses zero-shot and point adjustment comparisons. How would SPIE-AD perform on TimeseriesBench?
> > > > >
> > > > > Answer: TimeseriesBench is composed of datasets like Yahoo and UCR. It was pointed out in public comment as well as in “Current Time Series Anomaly Detection Benchmarks are Flawed and are Creating the Illusion of Progress” by Wu et al. IEEE TKDE 2023 that Yahoo dataset has mislabeling errors. Hence, we have tested SPIE-AD on the UCR database which has 250 real-world anomaly datasets. Instead of supplementary documents we have included the results in the main paper itself as a new benchmark in Table 2 and report results in Section 5.3 with new evaluation metric of event-based anomaly detection accuracy. We have also performed a sensitivity analysis SPIE-AD to varying window size on the UCR database.
> > > > >
> > > > > TimeSeriesBench makes an argument for using reduced length point adjustment, where the true positives are multiplied by a factor ln(k+e) where k is the length of the anomaly, and e is the severity of the anomaly. This is based on the hypothesis that longer anomalies are harder to detect and hence should have larger weight in the overall true positive computation. However, we have seen in our experiments that longer anomalies are actually easier to detect. The reason is a longer anomaly allows us a longer window to learn the underlying dynamics better. One of our major drawbacks is that we miss shorter anomalies that are less than 5 samples in the SMD, SMAP and MSL datasets. This is discussed in the new limitation section in the revised paper Section 6.
> > > > >
> > > > > Further, TimeSeriesBench argues that longer anomalies may be more critical. Hence the factor e is proportional to k. However, in U2 scenario this is not the case. In fact, U2 scenarios such as wrongful triggering of MCAS, or misidentification of objects by autonomous cars, or insulin cartridge failure are sudden failures that cause short term abrupt loss of control and safety. Hence, for U2 cases, short term anomalies are more important to capture.
> > > > >
> > > > > Reduced length point adjustments have the same problems as point adjustments itself where the true positives are inflated based on the length of the anomaly and a severity factor that considers longer anomalies being more severe which is not the case for U2 scenarios.

---

> > > > > > ### Author Response · Authors · 2024-12-03
> > > > > >
> > > > > > Dear reviewer,
> > > > > >
> > > > > > Hope you have gotten a chance to check the revised paper. Please let us know if you need any further clarifications.

---

### Public Comment · ~Eamonn_Keogh1 · 2024-11-24
**You test on SMD SMAP MSL. However it is increasingly understood that it is meaningless to test on these datasets**

You test on SMD SMAP MSL. However it is increasingly understood that it is meaningless to test on these datasets, because the ground truth is mislabeled, they are trivial, and have other flaws. See [a]

Perhaps I misunderstand, but isn't the Matrix Profile TSAD algorithm zero shot?  (and zero parameter?)
https://www.youtube.com/watch?v=vH4MzuaBeOQ&ab_channel=EamonnKeogh

[a]
https://www.dropbox.com/scl/fi/x6ie264xrfkl0nbdw1vtb/Irrational-Exuberance_why_most_TSAD_is_wrong.pdf?rlkey=16frcr2lo6ip5o6uf18qwdeud&dl=0
and
https://www.dropbox.com/scl/fi/cwduv5idkwx9ci328nfpy/Problems-with-Time-Series-Anomaly-Detection.pdf?rlkey=d9mnqw4tuayyjsplu0u1t7ugg&dl=0

---

> ### Author Response · Authors · 2024-11-25
> **SMD SMAP and MSL only used to compare with benchmarks and baselines**
>
> Thank you very much for your comment.
>
> ===
>
> You test on SMD SMAP MSL. However it is increasingly understood that it is meaningless to test on these datasets, because the ground truth is mislabeled, they are trivial, and have other flaws. See [a]
>
> ===
>
> Our main aim of the paper is detection of unknown unknowns (U2) which is a special case of zero shot MTAD where anomalous data is not available in training data. As such the existing benchmark MTAD datasets used in the baselines do not conform because in training data these datasets have anomalous data. This is actually similar to the semi-supervised task in the arxiv version of the Matrix Profile paper https://www.arxiv.org/pdf/2409.09298.
>
> However, to compare with baselines we still needed to execute our technique on benchmark anomaly detection datasets. This is why our main result in Table 3 does not include the SMD, SMAP and MSL datasets. It only uses the new datasets that were obtained from simulating real world systems such as unmanned autonomous vehicles, Autonomous insulin delivery systems, and F8 Cruiser aircraft. However, to evaluate our technique on standard real world datasets we have used SMD SMAP and MSL.
>
> ===
> To your point about the datasets being trivial.
> ===
>
> This is exactly what we also observed in our paper. If you look at Figure 1, you will see a significant difference in the distributions between normal and anomalous data. We show in our paper that a simple statistical method without any training, AnomalySimpleton actually performs on par or beats state-of-the-art anomaly detection approaches as shown in Table 4 of our paper.
>
> ===
> To the point about ground truth being mislabelled.
> ===
>
> We were actually unaware about mislabelling. May be that explains SPIE-AD having poorer performance than state of the art baseline techniques in the SMD, SMAP, and MSL datasets as seen in Table 4.
>
> ===
> Additional issues with inflated precision.
> ===
>
> We looked at the presentation that you posted in your comment. We agree with nearly every point that is discussed about the short comings of state-of-the-art anomaly detection technique. In this paper, we have highlighted the following which is in line with your presentation as well.
>
> a)	Usage of validation data from test
> Nearly all techniques set their validation data to be drawn from the test dataset. This is a basic error in ML principles and leads to data leakage, where the techniques are overfitted to the given test dataset. Validation data should always be taken from training data and test data should be left untouched. When we replace test data by train data for the validation set, the performance of state-of-the-art techniques reduce even further.
>
> b)	Use of point adjustment
> Nearly all state-of-the-art techniques use point adjustment (PA) to inflate precision.
> Kim et al AAAI 2022 (also referred to in our paper) showed that PA leads to over-estimation. In section 3.2 of Kim et al, they prove that random anomaly score assignment with point adjustment leads to high F1 score except for cases when anomalies are really small.
> We have also shown this in our untrained AnomalySimpleton approach which achieves high F1 score using PA.
>
> The problem with PA is the argument of precision. PA drastically inflates precision. In real world usage, the anomaly detection method will not have access to ground truth. Thus PA cannot be used. So the real precision of the method will be very very low. This indicates large number of false positives as compared to true positives. This leads to alarm fatigue. Consider this system being installed in a hospital. A nurse having alarm fatigue will most definitely ignore true positives also. So ultimately the anomaly detection method will be useless.
>
> ===
> To your point about Matrix Profile TSAD algorithm being zero shot
> ===
>
> From our limited understanding of the Matrix Profile method, it seems that the semi-supervised task in the paper is similar to zero shot MTAD. However, from our initial investigation it seems that Matrix Profile TSAD algorithm was initially developed for univariate time series and the multi-variate extension was only made public in September 2024. Very close to the submission deadline of ICLR. We were unfortunately unaware of the multivariate version of Matrix profile algorithm. We also only today got access to the code from the arxivs paper https://sites.google.com/view/mp4ad. We are attempting to run the method in our datasets. However, given the timeline and the need for responding to all reviewers we may not have enough time to include full results in our table. We will definitely discuss Matrix Profile as an alternate approach to zero shot MTAD in our revised paper.

---

### Author Response · Authors · 2024-12-02
**Summary of changes**

Dear Reviewers and Area Chair,

Based on the review comments we have made the following changes in our paper:

1) Added real world example of U2 scenarios and the properties of datasets induced by U2 scenarios.

2) Clarified the problem definition of anomaly detection in Figure 1

3) Added three new anomaly detection baselines OFA, TFAD and FITS

4) Added new real world benchmark dataset from UCR that has 250 real world timeseries with anomalies

5) Added computational complexity of SPIE-AD and compared the execution time of baselines with SPIE-AD

6) Added a new experiment to determine sensitivity of SPIE-AD to window size Figure 5

7) Expanded on the limitations of the approach in a limitation section.

After the revision there has been limited interaction with reviewers. Please let us know if you have any more feedback after the revisions.

---

### Meta-Review · Area_Chair_C2sC · 2024-12-20

**Metareview:**

The work aims to address a suspicious information leakage issue in multivariate time series anomaly detection, where anomaly detectors require validation/test data with anomaly samples to determine an effective threshold for the binary labeling. It proposes an approach based on sparse model recovery to tackle the issue.

As pointed out by the reviewers, the strengths of the work include: 1) the sparse model recovery-based method is new or theoretically sound (fJ9Q, V6nE, T8KH, sZLJ, awo8), 2) the proposed evaluation protocol is interesting and sounds more practical (fJ9Q, V6nE, T8KH, awo8), 3) the creation of synthetic evaluation datasets for avoiding the information leakage problem (T8KH)

The reviewers also identify a number of weaknesses: 1) some major paper clarity issues (fJ9Q, T8KH, sZLJ, awo8), 2) missing of comparison to some baselines, relevant benchmarks (fJ9Q, V6nE, T8KH, sZLJ, awo8), 3) the proposed method is not straightforward and its time complexity analysis is missing (V6nE), 4) lack of empirical justification on real-world datasets (V6nE), 4) the studied setting is unrealistic to some extent due to possible noise/anomaly contamination in real-life data (sZLJ), 5) lack of failure case analysis (sZLJ)

**Additional Comments On Reviewer Discussion:**

Five reviewers submitted their reviews for this paper, four of which participate in the discussion. The rebuttal helps clarify some of the confusions raised, address the time complexity analysis issue and the evaluation on real-world data. These result in an increased rating from Reviewer V6nE. However, the rebuttal does not convince the other four reviewers, with their original ratings [three weak rejects and one reject] unchanged, among which reviewer sZLJ did not engage in the discussion.

Overall, the work tackles an interesting and practical TSAD problem and helps create a setting that may avoid the suspicious information leakage problem in TSAD. The proposed method is considered innovative by the reviewers, but the empirical justification is still weak. Although the rebuttal helps address the empirical justification issue to some extent, the reviewers generally believe that the paper could be enhanced significantly with more iteration.

One issue raised by reviewer fJ9Q but not addressed by the authors is about the comparison to the emerging time series foundation models that have zero-shot generalization abilities. This could be an crucial issue. This is mainly because that although the paper claims and defines its own "zero-shot" TSAD problem, it is significantly different from the widely perceived "zero-shot" capability that emphasizes good performance on a target dataset without using any of its training data.

Considering all these factors, the work is not accepted to ICLR.

---

### Decision · Program_Chairs · 2025-01-22

Reject